# Test-Time Adaptation for Unsupervised Combinatorial Optimization

**Yiqiao Liao**                                                       *yil345@ucsd.edu*
*UC San Diego*

**Farinaz Koushanfar**                                               *farinaz@ucsd.edu*
*UC San Diego*

**Parinaz Naghizadeh**                                               *parinaz@ucsd.edu*
*UC San Diego*

**Reviewed on OpenReview:** *https://openreview.net/forum?id=VVyGfRp4fG*

## Abstract

Unsupervised neural combinatorial optimization (NCO) enables learning powerful solvers without access to ground-truth solutions. Existing approaches fall into two disjoint paradigms: models trained for generalization across instances, and instance-specific models optimized independently at test time. While the former are efficient during inference, they lack effective instance-wise adaptability; the latter are flexible but fail to exploit learned inductive structure and are prone to poor local optima. This motivates the central question of our work: how can we leverage the inductive bias learned through generalization while unlocking the flexibility required for effective instance-wise adaptation? We first identify a challenge in bridging these two paradigms: generalization-focused models often constitute poor warm starts for instance-wise optimization, potentially underperforming even randomly initialized models when fine-tuned at test time. To resolve this incompatibility, we propose TACO, a model-agnostic test-time adaptation framework that unifies and extends the two existing paradigms for unsupervised NCO. TACO applies strategic warm-starting to partially relax trained parameters while preserving inductive bias, enabling rapid and effective unsupervised adaptation. Crucially, compared to naively fine-tuning a trained generalizable model or optimizing an instance-specific model from scratch, TACO achieves better solution quality while incurring negligible additional computational cost. Experiments on the canonical problems of minimum vertex cover, maximum clique, maximum independent set, and max cut demonstrate the effectiveness and robustness of TACO across static, distribution-shifted, and dynamic combinatorial optimization problems, establishing it as a practical bridge between generalizable and instance-specific unsupervised NCO.

## 1 Introduction

Combinatorial optimization (CO) problems are central to many real-world applications, ranging from routing and scheduling to resource allocation and logistics (Papadimitriou & Steiglitz, 1982). These problems are notoriously hard to solve at scale due to their discrete and often non-convex structure. Neural combinatorial optimization (NCO) has emerged as a promising alternative to traditional solvers by learning solution heuristics directly from data (Joshi et al., 2019; 2022; Gasse et al., 2019; Hudson et al., 2022; Bello et al., 2017; Khalil et al., 2017; Li et al., 2024; 2023). Recent advances in unsupervised learning frameworks have enabled the training of powerful NCO solvers without requiring optimal or near-optimal solutions (Karalias & Loukas, 2020; Wang & Li, 2023; Toenshoff et al., 2021; Schuetz et al., 2022; Wang et al., 2022).

Two primary paradigms have emerged within unsupervised NCO: *generalization-focused* and *instance-based* methods. The first focuses on learning problem-specific heuristics from a diverse set of training instances, aim-

ing for strong *generalization* to unseen problem instances (Karalias & Loukas, 2020; Wang & Li, 2023). Once trained, these models are typically deployed to generate solutions in a single forward pass, with limited to no feedback or adaptation to the specific test instance. While this allows for efficient inference, it limits performance in scenarios involving distribution shifts or dynamic constraints, common conditions in real-world applications (Yang et al., 2012; Zhang et al., 2021). In contrast, the second paradigm focuses on *instance-specific* optimization, where a model is optimized independently for each test instance, aiming for instance-wise good solutions, without requiring access to a training dataset containing diverse problem structures (Schuetz et al., 2022; Ichikawa, 2024; Heydaribeni et al., 2024). As a result, this paradigm stays unaffected by distribution shifts and dynamic changes, but fails to leverage transferable structures and is potentially susceptible to becoming trapped in poor local optima during optimization (Wang & Li, 2023; Liao et al., 2025).

A natural idea is to unify these paradigms by adapting a trained generalizable model to each test instance. Perhaps surprisingly, we find that simply fine-tuning trained unsupervised NCO models at test time can perform *worse* than optimizing from random initialization, even under identical objectives and optimization budgets. This counterintuitive behavior reveals a fundamental incompatibility between distribution-level training and instance-level optimization in unsupervised NCO: the optimization landscape around trained parameters may not be conducive to rapid adaptation, potentially due to overfitting or local minima.

To address this challenge, we propose TACO (**T**est-time **A**daptation for unsupervised **C**ombinatorial **O**ptimization), a test-time adaptation framework that bridges generalization-focused and instance-specific unsupervised NCO. Our method is model-agnostic and complements existing generalization-based unsupervised NCO pipelines, offering plug-and-play integration. Experiments on minimum vertex cover, maximum clique, maximum independent set, and max cut with three NCO frameworks as backbones (EGN (Karalias & Loukas, 2020), Meta-EGN (Wang & Li, 2023), and ConsFormer (Xu et al., 2025)) under three different problem settings (static problems, distribution shifts, and dynamic environments) show that TACO achieves consistent improvement in solution quality with negligible additional computational overhead compared to either fine-tuning a trained generalizable model or optimizing an instance-specific model from scratch. We also compare TACO to a number of non-neural baselines, as well as Gurobi (Gurobi Optimization, LLC, 2024), a leading commercial optimization solver, and show that TACO attains at least comparable if not superior solution quality relative to the non-neural baselines, and has faster runtime than exact solvers like Gurobi on hard problem instances. These results establish TACO as a practical and efficient framework unifying generalizable and instance-specific unsupervised NCO.

## 2 Preliminaries

Let $G = (V, E)$ be an undirected graph, where $V$ is the set of nodes with $|V| = n$, and $E \subseteq V \times V$ is the set of edges. We define the solution to a CO problem on graph $G$ as a vector $x \in \mathcal{X}(G)$, where $\mathcal{X}(G) \subseteq \{0, 1\}^n$ denotes the feasible solution space over $G$, depending on the problem constraints. The general form of a CO problem can thus be written as:

$$\min_{x \in \mathcal{X}(G)} f(G, x),$$

where $f(G, x)$ is a problem-specific objective function, such as vertex cover size or negative clique size, and $\mathcal{X}(G)$ encodes constraints like covering or connectivity. We begin with an overview of the two existing paradigms for unsupervised NCO. While both paradigms employ unsupervised objectives, they follow fundamentally different optimization goals, with one trained for optimizing average-case performance and the other optimizing directly for a single instance.

### 2.1 Unsupervised NCO with generalization: Erdős Goes Neural (EGN) and Meta-EGN

**EGN.** To tackle CO problems in a label-free setting, Karalias & Loukas (2020) introduced EGN, a principled unsupervised learning approach inspired by Erdős' probabilistic method. Concretely, a graph neural network (GNN) $g_\theta$ is trained by minimizing the objective to map an input graph $G$ to a distribution $D = g_\theta(G)$ over binary vectors $x \in \{0, 1\}^n$. Each component $x_i$ is modeled as a Bernoulli random variable with probability $p_i = g_\theta(G)_i$, denoting the likelihood of including node $v_i$ in the solution. In the constrained setting, constraint

violations are penalized by augmenting the objective function:

$$\ell(D; G) := \mathbb{E}_{x \sim D}[f(G, x)] + \beta \cdot \mathbb{P}(x \notin \mathcal{X}(G)), \tag{1}$$

where $\beta \in \mathbb{R}_{>0}$ is a penalty parameter. Once trained, the learned distribution is used to decode a discrete solution via sequential decoding. This sequential process greedily fixes binary decisions $x_i$, one node at a time, so that the assignment of that node maintains or improves the expected objective. This ensures a deterministic and constraint-feasible binary solution.

**Meta-EGN.** While EGN learns generalizable heuristics from training data, it optimizes for averaged performance over the distribution of problem instances and may fail to provide high-quality solutions for individual test instances, especially under distribution shifts. To overcome this limitation, Wang & Li (2023) proposed Meta-EGN, a meta-learning extension of EGN designed to refine the model for improving instance-wise solutions. Inspired by model-agnostic meta-learning (MAML) (Finn et al., 2017), Meta-EGN views each training instance as a pseudo-test case. Instead of directly learning a solution-generating network, Meta-EGN seeks to learn a parameter initialization that can be quickly fine-tuned on unseen test instances. During inference, Meta-EGN either uses the pre-adapted model or performs gradient updates for further refinement.

Meta-EGN offers instance-wise adaptability by leveraging meta-learning *during training*, whereas our proposed method, TACO, improves adaptability *at test-time*. Moreover, Wang & Li (2023) evaluate Meta-EGN using only single-step gradient updates. In contrast, we conduct a systematic empirical analysis across a range of test-time update budgets. We show that our approach of test-time adaptation can outperform meta-learning-based adaptation, and that the performance of Meta-EGN can be further improved with negligible additional overhead when the two approaches are paired, indicating the orthogonality of the two approaches.

## 2.2 Unsupervised NCO with instance-specific optimization: PI-GNN

PI-GNN (Schuetz et al., 2022) is a general unsupervised framework for CO problems formulated as a quadratic unconstrained binary optimization (QUBO) (Lucas, 2014; Glover et al., 2022; Djidjev et al., 2018). Given an instance of a CO problem, PI-GNN learns a solution via $g_\theta(G)$. Since the input graph lacks node features, PI-GNN initializes learnable node embeddings randomly and passes them through a GNN. The model outputs a relaxed solution $x \in [0, 1]^n$ by optimizing a differentiable QUBO objective, followed by a rounding step to produce a valid binary solution. As PI-GNN applies a GNN to each problem instance individually and optimizes the corresponding QUBO objective, it operates in a fully training-data-free, instance-specific manner.

Prior works have shown that PI-GNN can perform worse than EGN and Meta-EGN on dense graphs (Wang & Li, 2023; Ichikawa, 2024). For this reason and consistency in baseline selections, we consider an instance-specific EGN variant that is optimized from scratch for each test instance. We discuss more recent extensions to PI-GNN in Section 6.

## 2.3 The challenge in bridging the two paradigms

A straightforward approach to combining generalizable and instance-specific unsupervised NCO would be to simply fine-tune a trained model on each test instance. However, we identify a failure mode: directly fine-tuning trained models at test time often underperforms instance-specific optimization from random initialization, even when using identical unsupervised objectives, learning rates, and update steps.

Figure 1 illustrates this phenomenon by comparing adaptation trajectories starting from trained parameters versus random initialization on the same test instances. Models initialized from trained parameters typically achieve better objective values in the earliest optimization steps, indicating that they encode useful inductive biases learned during distribution-level training. However, their performance either rapidly plateaus or is frequently overtaken by randomly initialized models, which, despite poorer initial performance, continue to improve over longer optimization horizons. This phenomenon suggests that the optimization landscape around trained parameters may be less conducive to rapid adaptation, potentially due to overfitting or local minima, motivating a more principled approach to both warm-starting and test-time updates.

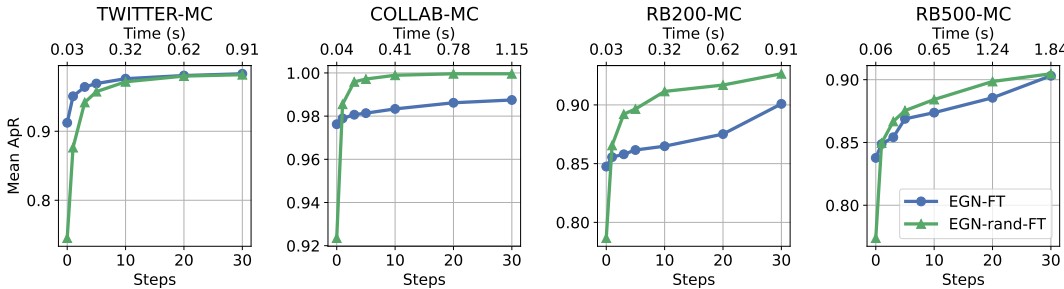

Figure 1: Performance (↑) of trained and randomly initialized (rand) EGN models with respect to the number of fine-tuning (FT) steps. Detailed setup is explained in Section 4.

## 3 Method

Unifying the strengths of generalization and instance-specific optimization in unsupervised NCO, our method, TACO (**T**est-time **A**daptation for unsupervised **C**ombinatorial **O**ptimization), builds upon a trained unsupervised NCO model and adapts it to individual test instances through a principled warm-starting and test-time optimization procedure. For each test instance, TACO performs a small number of unsupervised gradient updates starting from the trained parameters $\theta$, using the loss function of the same form employed during training, as defined in Equation 1. Crucially, instead of directly initializing from $\theta$, TACO applies a strategic initialization to preserve learned inductive biases while enabling effective adaptation. Specifically, the adapted parameters are initialized as:

$$\theta^* \leftarrow \lambda_{\text{shrink}} \cdot \theta + \lambda_{\text{perturb}} \cdot \epsilon,$$

where $0 < \lambda_{\text{shrink}} < 1$, $0 < \lambda_{\text{perturb}} < 1$, and $\epsilon \sim \mathcal{N}(\mathbf{0}, \boldsymbol{\Sigma})$. For models with specific parameter initialization techniques, the randomly initialized parameters will serve as the source of $\epsilon$. The shrink term contracts the trained parameters toward the origin, partially relaxing the strong biases imposed by training on a distribution of instances, while still preserving useful learned hypotheses. The perturbation term introduces controlled stochasticity, enabling exploration of nearby regions in the parameter space. Together, these effects move the initialization closer to a fresh start, encouraging learning and discovering alternative solutions, while retaining informative inductive biases learned during training. In practice, $\lambda_{\text{shrink}}$ and $\lambda_{\text{perturb}}$ can be selected using a validation set or the test instances available at hand. That said, in Section 4 we show that the performance of TACO is not overly sensitive to these hyperparameters.

Shrink and perturb (SP) was originally proposed in the *supervised learning* setting to address the generalization gap arising from naive warm-starting during training (Ash & Adams, 2020). In contrast, TACO adapts SP to a fundamentally different regime: adapting *unsupervised NCO* at test time. Through integration into the existing unsupervised NCO pipelines and extensive empirical analysis, we demonstrate that SP is highly effective in this new context. Specifically, we identify two main reasons for the effectiveness of SP in the unsupervised NCO context (as supported by our empirical analysis): (i) preserving the inductive biases encoded in the trained weights of NCO models (higher-quality solutions without any test-time optimization compared to fresh instance-specific models), and (ii) facilitating more favorable initializations for reaching better local optima faster, attained through both parameter scaling and stochastic perturbations (better solutions than the baselines across different numbers of update steps).

**Online TACO.** In standard TACO, each test instance is adapted independently by initializing from a fixed SP-transformed version of $\theta$. In many practical scenarios, test instances arrive sequentially and may share structural similarities. To exploit this, we introduce an online variant of TACO. Given a sequence of test instances $G_1, G_2, \ldots, G_M$, online TACO reuses the adapted parameters from instance $G_i$ as the basis for adapting to $G_{i+1}$. Concretely, we initialize from the most recent optimized parameters $\theta_i^*$, apply a fresh SP transformation (potentially with different $\lambda_{\text{shrink}}$ and $\lambda_{\text{perturb}}$), and then perform test-time optimization on $G_{i+1}$. This allows the model to accumulate and transfer knowledge across instances while maintaining adaptability. We illustrate standard and online TACO in Figure 2.

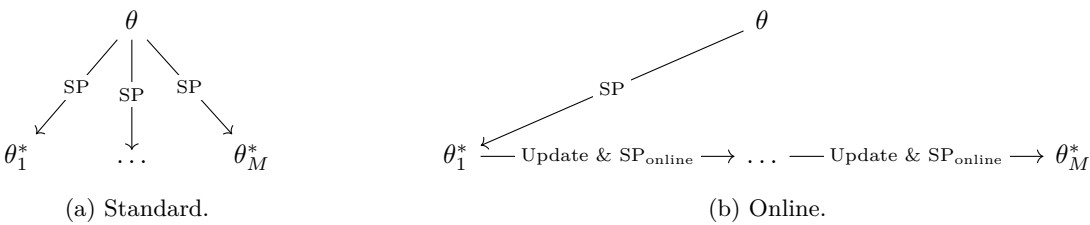

Figure 2: Two versions of TACO: standard vs. online.

# 4 Experiments

We empirically evaluate the effectiveness of TACO and online TACO on classical CO problems defined over graphs: minimum vertex cover (MVC), maximum clique (MC), and max cut (MaxCut). We consider three settings: (1) static graphs with fixed distribution, (2) distribution shifts in graph structures, and (3) dynamic graphs with temporal changes. We also include experiments on the maximum independent set (MIS) problem in Appendix B.5. Our code is available at `https://github.com/yl489/TTA4UCO`.

## 4.1 Datasets

**Static problems.** For the static setting, we employed real-world and synthetically generated graphs used in previous works (Karalias & Loukas, 2020; Karalias et al., 2022; Wang & Li, 2023; Sanokowski et al., 2024). The real-world datasets include Twitter (Leskovec & Krevl, 2014) and COLLAB (Yanardag & Vishwanathan, 2015), which represent social and collaboration networks, respectively. Additionally, we generated synthetic graphs using the RB model (Xu et al., 2007), producing two datasets: RB200 and RB500, with approximately 200 and 500 nodes per graph. Following Wang & Li (2023), we sampled the RB model parameter $p \in [0.3, 1]$ uniformly when generating the training and validation sets and fixed $p = 0.25$ for the test set to generate hard instances. For Twitter and COLLAB, we used a standard 60-20-20 train/validation/test split. For RB200 and RB500, we generated 2000 graphs for training, 100 graphs for validation, and 100 graphs for testing.

**Distribution shift.** To assess performance under distribution shift, we trained our models on the Twitter dataset and evaluated them on the RB200 test set, similar to Wang & Li (2023). This setup introduces a significant structural shift from real-world social graphs to synthetic rule-based graphs.

**Dynamic problems.** For the dynamic setting, we considered discrete-time dynamic graphs where a stream of graph snapshots is observed sequentially. Models were trained on static Twitter graphs and evaluated on two dynamic datasets: Twitter Tennis UO (Béres et al., 2018), a dynamic Twitter mention graph, for the MVC experiments, and COVID-19 England (Panagopoulos et al., 2021), a dynamic mobility graph, for the MC experiments. We took the top 150 popular nodes of Twitter Tennis UO for each snapshot, resulting in changes in both the node set $V$ and the edge set $E$. For COVID-19 England, the node set $V$ remains the same across all snapshots, and only the edge set $E$ changes over time. Both datasets are available in the PyTorch Geometric Temporal library (Rozemberczki et al., 2021). We selected Twitter Tennis UO for the MVC experiments only, since the clique sizes are in the range of 2 to 5, making performance comparison less meaningful. Similarly, COVID-19 England snapshots have minimum vertex covers almost equal to the node set, so we used them for the MC experiments only.

## 4.2 Implementation details and baselines

Our EGN and Meta-EGN backbone models are the same as the ones in prior works (Karalias & Loukas, 2020; Wang & Li, 2023), consisting of four Graph Isomorphism Network (Xu et al., 2019) layers. We used the Adam optimizer (Kingma & Ba, 2015) for training and tuning all models. For evaluation, we obtained the ground truth for each graph (snapshot) by solving the corresponding CO problem using the Gurobi solver (Gurobi Optimization, LLC, 2024) and report the mean approximation ratio (ApR) as the primary metric. Formally, ApR $= f(G, x)/f(G, x^*)$, where $x^*$ is the optimal solution. Additional implementation

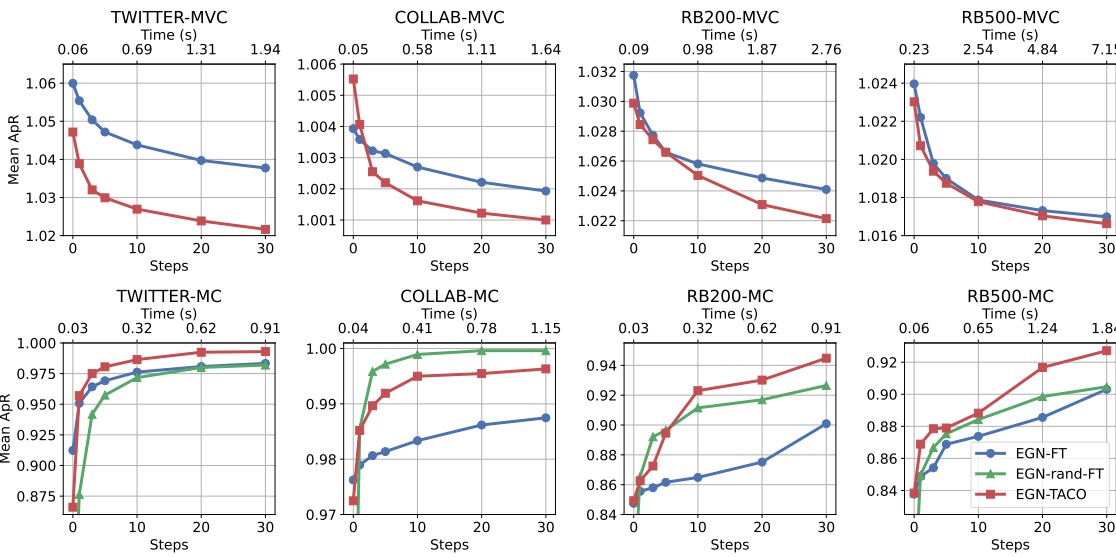

Figure 3: Mean ApR of methods using EGN as the backbone on static MVC (↓) and MC (↑) problems with respect to the number of update steps. "FT" stands for fine-tuning; "rand" means models are freshly initialized. The wall clock time factors in the decoding operations. Subplots not showing results of "EGN-rand-FT" are zoomed in for better illustration (i.e., freshly initialized models perform much worse). Figure 5 in Appendix B.1 shows all results.

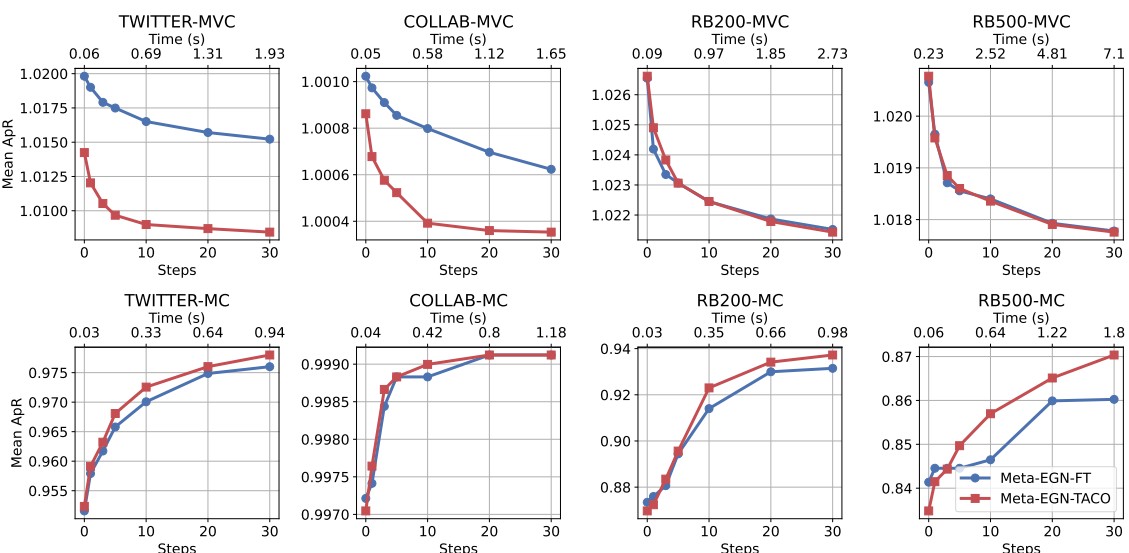

Figure 4: Mean ApR of methods using Meta-EGN as the backbone on static MVC (↓) and MC (↑) problems with respect to the number of update steps. "FT" stands for fine-tuning. The wall clock time factors in the decoding operations.

details, including the exact loss functions used for training and tuning all models, as well as the full set of optimizer, loss, and SP hyperparameters, are provided in Appendix A.

For baselines, we include fine-tuning trained and freshly initialized EGN models. We also consider an online fine-tuning variant, analogous to online TACO, to ensure fair comparisons. We apply a number of unsupervised gradient updates and compare the best solutions achieved so far by the baseline methods and TACO. Since EGN and Meta-EGN take a random one-hot vector of the nodes as the input, we examined the performance of the baselines and TACO with different numbers of random input initializations (seeds) in our experiments. Consistent with Karalias & Loukas (2020) and Wang & Li (2023), we take the seed leading to the best solution as the final output when using multiple input initializations.

Table 1: Mean ApR ($\downarrow$) and seconds per graph of all methods on MVC. All results are from 30 update steps. "FT" stands for fine-tuning; "Accurate (8)" means 8 seeds were used. The best solutions are in **bold**, and the cases where EGN + TACO outperform Meta-EGN-FT(-Online) are in gray.

| TWITTER | | | |
|---|---|---|---|
| Method | Fast (1) | Balanced (4) | Accurate (8) |
| EGN | $1.09349_{\pm 0.05062}(0.02)$ | $1.05996_{\pm 0.03552}(0.06)$ | $1.04928_{\pm 0.02902}(0.13)$ |
| EGN-FT | $1.06311_{\pm 0.03905}(0.48)$ | $1.03775_{\pm 0.02640}(1.94)$ | $1.03026_{\pm 0.02123}(3.88)$ |
| EGN-FT-Online | $1.04482_{\pm 0.03484}(0.50)$ | $1.02434_{\pm 0.02197}(1.95)$ | $1.01865_{\pm 0.01669}(3.94)$ |
| EGN-TACO | $1.03826_{\pm 0.03473}(0.49)$ | $1.02165_{\pm 0.02221}(1.95)$ | $1.01668_{\pm 0.01636}(3.90)$ |
| EGN-TACO-Online | $\mathbf{1.03082}_{\pm \mathbf{0.03329}}(\mathbf{0.50})$ | $\mathbf{1.01997}_{\pm \mathbf{0.02094}}(\mathbf{1.99})$ | $\mathbf{1.01357}_{\pm \mathbf{0.01417}}(\mathbf{3.96})$ |
| Meta-EGN | $1.02998_{\pm 0.02016}(0.02)$ | $1.01981_{\pm 0.01579}(0.06)$ | $1.01749_{\pm 0.01536}(0.12)$ |
| Meta-EGN-FT | $1.02304_{\pm 0.01693}(0.48)$ | $1.01522_{\pm 0.01392}(1.93)$ | $1.01325_{\pm 0.01283}(3.87)$ |
| Meta-EGN-FT-Online | $1.02454_{\pm 0.01737}(0.49)$ | $1.01430_{\pm 0.01312}(1.95)$ | $1.01238_{\pm 0.01200}(3.90)$ |
| Meta-EGN-TACO | $\mathbf{1.01472}_{\pm \mathbf{0.01426}}(\mathbf{0.49})$ | $\mathbf{1.00844}_{\pm \mathbf{0.01004}}(\mathbf{1.95})$ | $\mathbf{1.00728}_{\pm \mathbf{0.00922}}(\mathbf{3.91})$ |
| Meta-EGN-TACO-Online | $1.02947_{\pm 0.03407}(0.50)$ | $1.01634_{\pm 0.01536}(1.96)$ | $1.01319_{\pm 0.01413}(3.93)$ |

| COLLAB | | | |
|---|---|---|---|
| Method | Fast (1) | Balanced (4) | Accurate (8) |
| EGN | $1.01197_{\pm 0.03309}(0.01)$ | $1.00393_{\pm 0.01541}(0.05)$ | $1.00186_{\pm 0.00738}(0.11)$ |
| EGN-FT | $1.00906_{\pm 0.02962}(0.41)$ | $1.00193_{\pm 0.00927}(1.64)$ | $1.00071_{\pm 0.00442}(3.28)$ |
| EGN-FT-Online | $1.00761_{\pm 0.02794}(0.42)$ | $1.00131_{\pm 0.00792}(1.65)$ | $1.00036_{\pm 0.00304}(3.29)$ |
| EGN-TACO | $1.00772_{\pm 0.02808}(0.41)$ | $1.00100_{\pm 0.00618}(1.66)$ | $1.00040_{\pm 0.00341}(3.32)$ |
| EGN-TACO-Online | $\mathbf{1.00206}_{\pm \mathbf{0.00942}}(\mathbf{0.42})$ | $\mathbf{1.00042}_{\pm \mathbf{0.00369}}(\mathbf{1.65})$ | $\mathbf{1.00017}_{\pm \mathbf{0.00216}}(\mathbf{3.33})$ |
| Meta-EGN | $1.00392_{\pm 0.01252}(0.01)$ | $1.00102_{\pm 0.00580}(0.05)$ | $1.00073_{\pm 0.00455}(0.11)$ |
| Meta-EGN-FT | $1.00214_{\pm 0.00826}(0.41)$ | $1.00062_{\pm 0.00446}(1.65)$ | $1.00053_{\pm 0.00425}(3.29)$ |
| Meta-EGN-FT-Online | $1.00173_{\pm 0.00819}(0.41)$ | $1.00052_{\pm 0.00429}(1.64)$ | $1.00031_{\pm 0.00337}(3.29)$ |
| Meta-EGN-TACO | $\mathbf{1.00148}_{\pm \mathbf{0.00641}}(\mathbf{0.41})$ | $\mathbf{1.00035}_{\pm \mathbf{0.00304}}(\mathbf{1.66})$ | $\mathbf{1.00027}_{\pm \mathbf{0.00268}}(\mathbf{3.31})$ |
| Meta-EGN-TACO-Online | $1.00477_{\pm 0.03821}(0.42)$ | $1.00425_{\pm 0.04920}(1.66)$ | $1.00602_{\pm 0.05284}(3.32)$ |

| RB200 | | | |
|---|---|---|---|
| Method | Fast (1) | Balanced (4) | Accurate (8) |
| EGN | $1.03982_{\pm 0.01087}(0.02)$ | $1.03175_{\pm 0.00561}(0.09)$ | $1.02940_{\pm 0.00483}(0.18)$ |
| EGN-FT | $1.02878_{\pm 0.00565}(0.69)$ | $1.02409_{\pm 0.00428}(2.76)$ | $1.02243_{\pm 0.00495}(5.51)$ |
| EGN-FT-Online | $1.03074_{\pm 0.00648}(0.70)$ | $1.02395_{\pm 0.00454}(2.75)$ | $1.02125_{\pm 0.00473}(5.46)$ |
| EGN-TACO | $1.02758_{\pm 0.00530}(0.69)$ | $1.02214_{\pm 0.00462}(2.76)$ | $1.02052_{\pm 0.00420}(5.52)$ |
| EGN-TACO-Online | $\mathbf{1.02678}_{\pm \mathbf{0.00631}}(\mathbf{0.70})$ | $\mathbf{1.02168}_{\pm \mathbf{0.00469}}(\mathbf{2.75})$ | $\mathbf{1.01930}_{\pm \mathbf{0.00461}}(\mathbf{5.55})$ |
| Meta-EGN | $1.03394_{\pm 0.00746}(0.02)$ | $1.02655_{\pm 0.00496}(0.09)$ | $1.02502_{\pm 0.00478}(0.18)$ |
| Meta-EGN-FT | $1.02622_{\pm 0.00564}(0.68)$ | $1.02152_{\pm 0.00429}(2.73)$ | $1.01994_{\pm 0.00429}(5.47)$ |
| Meta-EGN-FT-Online | $1.02778_{\pm 0.00745}(0.68)$ | $1.02121_{\pm 0.00587}(2.73)$ | $1.02009_{\pm 0.00465}(5.54)$ |
| Meta-EGN-TACO | $\mathbf{1.02614}_{\pm \mathbf{0.00535}}(\mathbf{0.69})$ | $1.02143_{\pm 0.00505}(2.75)$ | $1.01979_{\pm 0.00474}(5.49)$ |
| Meta-EGN-TACO-Online | $1.03018_{\pm 0.01154}(0.69)$ | $\mathbf{1.02030}_{\pm \mathbf{0.00551}}(\mathbf{2.78})$ | $\mathbf{1.01903}_{\pm \mathbf{0.00497}}(\mathbf{5.51})$ |

| RB500 | | | |
|---|---|---|---|
| Method | Fast (1) | Balanced (4) | Accurate (8) |
| EGN | $1.02837_{\pm 0.00638}(0.06)$ | $1.02396_{\pm 0.00207}(0.23)$ | $1.02308_{\pm 0.00190}(0.46)$ |
| EGN-FT | $1.01951_{\pm 0.00284}(1.79)$ | $1.01699_{\pm 0.00229}(7.15)$ | $1.01630_{\pm 0.00208}(14.30)$ |
| EGN-FT-Online | $1.01846_{\pm 0.00251}(1.77)$ | $1.01653_{\pm 0.00226}(7.10)$ | $1.01518_{\pm 0.00197}(14.24)$ |
| EGN-TACO | $1.01878_{\pm 0.00280}(1.78)$ | $1.01663_{\pm 0.00230}(7.12)$ | $1.01577_{\pm 0.00214}(14.24)$ |
| EGN-TACO-Online | $\mathbf{1.01749}_{\pm \mathbf{0.00258}}(\mathbf{1.77})$ | $\mathbf{1.01540}_{\pm \mathbf{0.00202}}(\mathbf{7.12})$ | $\mathbf{1.01512}_{\pm \mathbf{0.00209}}(\mathbf{14.25})$ |
| Meta-EGN | $1.02328_{\pm 0.00302}(0.06)$ | $1.02065_{\pm 0.00226}(0.23)$ | $1.01983_{\pm 0.00229}(0.46)$ |
| Meta-EGN-FT | $1.01976_{\pm 0.00272}(1.78)$ | $1.01778_{\pm 0.00213}(7.10)$ | $1.01698_{\pm 0.00214}(14.21)$ |
| Meta-EGN-FT-Online | $1.01866_{\pm 0.00315}(1.78)$ | $1.01591_{\pm 0.00256}(7.07)$ | $1.01478_{\pm 0.00222}(14.20)$ |
| Meta-EGN-TACO | $1.01959_{\pm 0.00279}(1.78)$ | $1.01776_{\pm 0.00219}(7.11)$ | $1.01700_{\pm 0.00211}(14.23)$ |
| Meta-EGN-TACO-Online | $\mathbf{1.01566}_{\pm \mathbf{0.00360}}(\mathbf{1.77})$ | $\mathbf{1.01356}_{\pm \mathbf{0.00294}}(\mathbf{7.08})$ | $\mathbf{1.01213}_{\pm \mathbf{0.00273}}(\mathbf{14.23})$ |

## 4.3 Empirical results

**Static problems.** We begin by evaluating the compatibility of TACO with EGN and Meta-EGN. Figures 3 and 4 report the mean ApR as a function of the number of gradient update steps across all datasets and tasks, using EGN and Meta-EGN as backbones, respectively. All results are obtained by setting the number of seeds to 4 for all models. Compared to the baseline methods, where we update the parameters of a trained model or a freshly initialized model, TACO consistently achieves superior performance across a wide range

Table 2: Mean ApR ($\uparrow$) and seconds per graph of all methods on MC. All results are from 30 update steps. "FT" stands for fine-tuning; "Accurate (8)" means 8 seeds were used. The best solutions are in **bold**, and the cases where EGN + TACO outperform Meta-EGN-FT(-Online) are in gray.

| TWITTER | | | |
|---|---|---|---|
| Method | Fast (1) | Balanced (4) | Accurate (8) |
| EGN | $0.73856_{\pm 0.25477}(0.01)$ | $0.91233_{\pm 0.12556}(0.03)$ | $0.95073_{\pm 0.08131}(0.06)$ |
| EGN-FT | $0.93744_{\pm 0.13083}(0.23)$ | $0.98332_{\pm 0.06204}(0.91)$ | $0.99151_{\pm 0.04802}(1.83)$ |
| EGN-FT-Online | $0.94610_{\pm 0.12316}(0.24)$ | $0.98182_{\pm 0.06225}(0.94)$ | $0.98672_{\pm 0.04891}(1.84)$ |
| EGN-TACO | $\mathbf{0.95419}_{\pm \mathbf{0.10578}}(\mathbf{0.23})$ | $\mathbf{0.99295}_{\pm \mathbf{0.02656}}(\mathbf{0.93})$ | $\mathbf{0.99766}_{\pm \mathbf{0.01310}}(\mathbf{1.86})$ |
| EGN-TACO-Online | $0.95119_{\pm 0.10600}(0.24)$ | $0.98400_{\pm 0.05507}(0.95)$ | $0.99277_{\pm 0.03325}(1.90)$ |
| Meta-EGN | $0.91078_{\pm 0.12812}(0.01)$ | $0.95158_{\pm 0.09174}(0.03)$ | $0.96538_{\pm 0.07509}(0.06)$ |
| Meta-EGN-FT | $\mathbf{0.94930}_{\pm \mathbf{0.09735}}(\mathbf{0.24})$ | $0.97601_{\pm 0.06939}(0.94)$ | $0.98749_{\pm 0.05144}(1.89)$ |
| Meta-EGN-FT-Online | $0.94836_{\pm 0.09920}(0.24)$ | $0.97430_{\pm 0.06951}(0.94)$ | $0.98253_{\pm 0.05748}(1.85)$ |
| Meta-EGN-TACO | $0.94783_{\pm 0.10246}(0.24)$ | $0.97799_{\pm 0.06808}(0.95)$ | $0.98663_{\pm 0.05393}(1.89)$ |
| Meta-EGN-TACO-Online | $0.94625_{\pm 0.11002}(0.24)$ | $\mathbf{0.98244}_{\pm \mathbf{0.05885}}(\mathbf{0.94})$ | $\mathbf{0.99061}_{\pm \mathbf{0.03641}}(\mathbf{1.89})$ |

| COLLAB | | | |
|---|---|---|---|
| Method | Fast (1) | Balanced (4) | Accurate (8) |
| EGN | $0.84956_{\pm 0.29397}(0.01)$ | $0.97625_{\pm 0.10808}(0.04)$ | $0.99663_{\pm 0.02481}(0.07)$ |
| EGN-FT | $0.90862_{\pm 0.22514}(0.29)$ | $0.98747_{\pm 0.07870}(1.15)$ | $0.99916_{\pm 0.01238}(2.30)$ |
| EGN-FT-Online | $0.98019_{\pm 0.09857}(0.29)$ | $0.99930_{\pm 0.01114}(1.19)$ | $0.99969_{\pm 0.00693}(2.37)$ |
| EGN-TACO | $0.95378_{\pm 0.15099}(0.30)$ | $0.99631_{\pm 0.03368}(1.18)$ | $0.99976_{\pm 0.00580}(2.36)$ |
| EGN-TACO-Online | $\mathbf{0.98217}_{\pm \mathbf{0.09103}}(\mathbf{0.30})$ | $\mathbf{0.99937}_{\pm \mathbf{0.00931}}(\mathbf{1.20})$ | $\mathbf{1.00000}_{\pm \mathbf{0.00000}}(\mathbf{2.40})$ |
| Meta-EGN | $0.98965_{\pm 0.05931}(0.01)$ | $0.99721_{\pm 0.02583}(0.04)$ | $0.99813_{\pm 0.01932}(0.08)$ |
| Meta-EGN-FT | $0.99546_{\pm 0.02957}(0.29)$ | $0.99912_{\pm 0.01169}(1.18)$ | $0.99966_{\pm 0.00647}(2.35)$ |
| Meta-EGN-FT-Online | $0.99381_{\pm 0.03670}(0.29)$ | $0.99894_{\pm 0.01298}(1.19)$ | $0.99961_{\pm 0.00735}(2.37)$ |
| Meta-EGN-TACO | $0.99526_{\pm 0.03094}(0.30)$ | $0.99912_{\pm 0.01169}(1.19)$ | $0.99980_{\pm 0.00464}(2.38)$ |
| Meta-EGN-TACO-Online | $\mathbf{0.99787}_{\pm \mathbf{0.01923}}(\mathbf{0.30})$ | $\mathbf{0.99978}_{\pm \mathbf{0.00516}}(\mathbf{1.19})$ | $\mathbf{0.99990}_{\pm \mathbf{0.00301}}(\mathbf{2.41})$ |

| RB200 | | | |
|---|---|---|---|
| Method | Fast (1) | Balanced (4) | Accurate (8) |
| EGN | $0.76354_{\pm 0.14769}(0.01)$ | $0.84750_{\pm 0.14285}(0.03)$ | $0.91045_{\pm 0.11979}(0.06)$ |
| EGN-FT | $0.80304_{\pm 0.15676}(0.23)$ | $0.90088_{\pm 0.13542}(0.91)$ | $0.95659_{\pm 0.09698}(1.83)$ |
| EGN-FT-Online | $0.88725_{\pm 0.14130}(0.24)$ | $0.96475_{\pm 0.07824}(0.95)$ | $\mathbf{0.99455}_{\pm \mathbf{0.02159}}(\mathbf{1.90})$ |
| EGN-TACO | $0.84914_{\pm 0.15928}(0.23)$ | $0.94480_{\pm 0.10703}(0.94)$ | $0.98094_{\pm 0.05496}(1.88)$ |
| EGN-TACO-Online | $\mathbf{0.90714}_{\pm \mathbf{0.12065}}(\mathbf{0.24})$ | $\mathbf{0.97687}_{\pm \mathbf{0.06840}}(\mathbf{0.98})$ | $0.98465_{\pm 0.05296}(1.88)$ |
| Meta-EGN | $0.77893_{\pm 0.18384}(0.01)$ | $0.87345_{\pm 0.13786}(0.03)$ | $0.89779_{\pm 0.11543}(0.06)$ |
| Meta-EGN-FT | $0.87840_{\pm 0.14259}(0.24)$ | $0.93146_{\pm 0.10693}(0.98)$ | $0.94966_{\pm 0.09027}(1.95)$ |
| Meta-EGN-FT-Online | $0.90159_{\pm 0.12417}(0.25)$ | $0.93530_{\pm 0.09976}(0.98)$ | $0.95546_{\pm 0.07421}(1.98)$ |
| Meta-EGN-TACO | $0.88296_{\pm 0.14071}(0.25)$ | $0.93723_{\pm 0.10152}(1.00)$ | $0.94984_{\pm 0.09018}(1.99)$ |
| Meta-EGN-TACO-Online | $\mathbf{0.91264}_{\pm \mathbf{0.11546}}(\mathbf{0.25})$ | $\mathbf{0.96069}_{\pm \mathbf{0.07413}}(\mathbf{1.01})$ | $\mathbf{0.97881}_{\pm \mathbf{0.05873}}(\mathbf{1.93})$ |

| RB500 | | | |
|---|---|---|---|
| Method | Fast (1) | Balanced (4) | Accurate (8) |
| EGN | $0.80616_{\pm 0.20549}(0.01)$ | $0.83771_{\pm 0.19788}(0.06)$ | $0.88334_{\pm 0.17438}(0.12)$ |
| EGN-FT | $0.82545_{\pm 0.19791}(0.46)$ | $0.90312_{\pm 0.16580}(1.84)$ | $0.95306_{\pm 0.11117}(3.68)$ |
| EGN-FT-Online | $\mathbf{0.87644}_{\pm \mathbf{0.18240}}(\mathbf{0.49})$ | $0.93703_{\pm 0.14093}(1.95)$ | $0.98121_{\pm 0.06640}(3.82)$ |
| EGN-TACO | $0.83504_{\pm 0.19576}(0.47)$ | $0.92717_{\pm 0.14329}(1.88)$ | $0.97101_{\pm 0.07950}(3.75)$ |
| EGN-TACO-Online | $0.86607_{\pm 0.18805}(0.48)$ | $\mathbf{0.94717}_{\pm \mathbf{0.13002}}(\mathbf{1.98})$ | $\mathbf{0.98684}_{\pm \mathbf{0.04781}}(\mathbf{3.88})$ |
| Meta-EGN | $0.79767_{\pm 0.20632}(0.01)$ | $0.84136_{\pm 0.19527}(0.06)$ | $0.88511_{\pm 0.17458}(0.12)$ |
| Meta-EGN-FT | $0.80896_{\pm 0.20270}(0.45)$ | $0.86025_{\pm 0.18965}(1.80)$ | $0.91018_{\pm 0.15735}(3.61)$ |
| Meta-EGN-FT-Online | $0.82832_{\pm 0.19989}(0.47)$ | $0.90237_{\pm 0.17033}(1.89)$ | $0.95514_{\pm 0.11758}(3.75)$ |
| Meta-EGN-TACO | $0.82316_{\pm 0.19892}(0.45)$ | $0.87036_{\pm 0.18832}(1.81)$ | $0.91610_{\pm 0.15453}(3.63)$ |
| Meta-EGN-TACO-Online | $\mathbf{0.86786}_{\pm \mathbf{0.18423}}(\mathbf{0.48})$ | $\mathbf{0.91854}_{\pm \mathbf{0.15807}}(\mathbf{1.89})$ | $\mathbf{0.96691}_{\pm \mathbf{0.09560}}(\mathbf{3.78})$ |

of update budgets. More importantly, within 10 update steps, TACO can achieve solutions unattainable by naively fine-tuning trained models for 30 steps, and TACO can outperform optimizing freshly initialized models when fine-tuning trained models falls short. As mentioned earlier, even though Meta-EGN enables instance-wise adaptability, its adaptability can still be improved when paired with TACO.

Next, we assess the robustness of all methods under varying numbers of random seeds. Tables 1 and 2 summarize these results. Models enhanced with TACO consistently achieve the best performance across

Table 3: Mean ApR ($\downarrow$) and seconds per graph by TACO-enhanced models and non-neural baselines on static MVC instances. (r) indicates the results are reported in Wang & Li (2023). The quality-runtime tradeoff of the TACO-enhanced models is detailed in Table 1.

| Method | Twitter | COLLAB | RB200 | RB500 |
|---|---|---|---|---|
| EGN-TACO | $1.02165_{\pm0.02221}(1.95)$ | $1.00100_{\pm0.00618}(1.66)$ | $1.02052_{\pm0.00420}(5.52)$ | $1.01577_{\pm0.00214}(14.24)$ |
| EGN-TACO-Online | $1.01997_{\pm0.02094}(1.99)$ | $1.00042_{\pm0.00369}(1.65)$ | $1.01930_{\pm0.00461}(5.55)$ | $1.01512_{\pm0.00209}(14.25)$ |
| Meta-EGN-TACO | $1.00844_{\pm0.01004}(1.95)$ | $1.00035_{\pm0.00304}(1.66)$ | $1.01979_{\pm0.00474}(5.49)$ | $1.01700_{\pm0.00211}(14.23)$ |
| Meta-EGN-TACO-Online | $1.01634_{\pm0.01536}(1.96)$ | $1.00425_{\pm0.04920}(1.66)$ | $1.01903_{\pm0.00497}(5.51)$ | $1.01213_{\pm0.00273}(14.23)$ |
| Degree-based Greedy (r) | $1.014_{\pm0.014}(1.95)$ | $1.209_{\pm0.198}(1.79)$ | $1.124_{\pm0.002}(5.02)$ | $1.062_{\pm0.005}(15.59)$ |
| Gurobi | $1.000\ (0.03)$ | $1.000\ (0.06)$ | $1.000\ (0.84)$ | $1.000\ (132.01)$ |

Table 4: Mean ApR ($\uparrow$) and seconds per graph by TACO-enhanced models and non-neural baselines on static MC instances. (r) indicates the results are reported in Sun et al. (2023), Ichikawa & Arai (2025), or Wang & Li (2023). The quality-runtime tradeoff of the TACO-enhanced models is detailed in Table 2.

| Method | Twitter | COLLAB | RB200 | RB500 |
|---|---|---|---|---|
| EGN-TACO | $0.99766_{\pm0.01310}(1.86)$ | $0.99976_{\pm0.00580}(2.36)$ | $0.98094_{\pm0.05496}(1.88)$ | $0.97101_{\pm0.07950}(3.75)$ |
| EGN-TACO-Online | $0.99277_{\pm0.03325}(1.90)$ | $1.00000_{\pm0.00000}(2.40)$ | $0.98465_{\pm0.05296}(1.88)$ | $0.98684_{\pm0.04781}(3.88)$ |
| Meta-EGN-TACO | $0.98663_{\pm0.05393}(1.89)$ | $0.99980_{\pm0.00464}(2.38)$ | $0.94984_{\pm0.09018}(1.99)$ | $0.91610_{\pm0.15453}(3.63)$ |
| Meta-EGN-TACO-Online | $0.99061_{\pm0.03641}(1.89)$ | $0.99990_{\pm0.00301}(2.41)$ | $0.97881_{\pm0.05873}(1.93)$ | $0.96691_{\pm0.09560}(3.78)$ |
| iSCO (r) | $1.000_{\pm0.000}(1.67)$ | - | $0.857_{\pm0.062}(1.67)$ | - |
| PQQA (r) | $1.000_{\pm0.000}(0.53)$ | - | $0.868_{\pm0.061}(1.51)$ | - |
| Toenshoff-Greedy (r) | $0.917_{\pm0.126}(0.08)$ | $0.969_{\pm0.087}(0.06)$ | $0.786_{\pm0.195}(2.25)$ | $0.793_{\pm0.202}(2.38)$ |
| Gurobi | $1.000\ (0.11)$ | $1.000\ (0.04)$ | $1.000\ (0.76)$ | $1.000\ (35.40)$ |

almost all settings, with the online version potentially offering additional gains. We also include the mean ApR of EGN with 256 seeds in Table 14 in Appendix B.2. In nearly all cases, TACO-enhanced models discover better solutions in a comparable amount of wall-clock time, except for the MC task on RB200 and RB500. These exceptions may be attributed to the nature of RB200 and RB500. Since the cliques are generated deliberately, and the random one-hot input vector can be interpreted as an initial guess, in the extreme setting, exhaustive exploration of initial guesses leads to strong performance. Nevertheless, we note that our goal is not to beat EGN and Meta-EGN with a large number of runs in comparable or less runtime and many fewer runs. TACO is orthogonal to the number of seeds, and the different runs of EGN and Meta-EGN can be executed in parallel, with each run paired with TACO, so the runtime does not scale linearly with the number of runs. Although TACO is a model-agnostic framework to enhance unsupervised NCO models and not explicitly designed to compete with MAML in NCO, EGN with TACO finds better solutions faster than standard fine-tuning and can surpass fine-tuned Meta-EGN in about half of the cases in the MVC experiments and nearly all cases in the MC experiments, as highlighted in Tables 1 and 2.

**Comparison with non-neural methods on static problems.** We further compare the TACO-enhanced models with representative non-neural baselines. Specifically, we consider the degree-based greedy heuristic and Toenshoff-Greedy (Toenshoff et al., 2021), following Wang & Li (2023) and Sanokowski et al. (2023). For the MC problem, we additionally include two sampling-based methods, iSCO (Sun et al., 2023) and PQQA (Ichikawa & Arai, 2025). As shown in Tables 3 and 4, TACO-enhanced models substantially outperform the greedy heuristics and achieve comparable or superior solution quality relative to the sampling-based approaches. Notably, on more challenging instances (RB200), the TACO-enhanced models greatly surpass the sampling-based methods. Furthermore, on RB500, the hardest instances considered in our experiments, TACO-enhanced models are able to obtain high-quality solutions within within 15 seconds, whereas Gurobi may require several minutes to converge. These results highlight the promise of unsupervised neural combinatorial optimization augmented with TACO for solving large-scale and difficult problem instances.

**Comparison against state-of-the-art NCO performance.** We also compare our method against the state-of-the-art NCO approach COExpander (Ma et al., 2025). Tables 5 and 6 summarize the performance of different COExpander configurations alongside TACO-enhanced Meta-EGN. On the MVC task, Meta-

Table 5: Mean ApR ($\downarrow$) and seconds per graph of COExpander and TACO-enhanced Meta-EGN on MVC.

| Method | Twitter | COLLAB |
|---|---|---|
| COExpander (S=4, D=20, I=1) | $1.00334_{\pm 0.00759}(0.04)$ | $1.00091_{\pm 0.00464}(0.03)$ |
| COExpander (S=8, D=20, I=1) | $1.00367_{\pm 0.00788}(0.04)$ | $1.00079_{\pm 0.00404}(0.04)$ |
| COExpander (S=1, D=5, I=20) | $1.00499_{\pm 0.00916}(0.33)$ | $1.00029_{\pm 0.00259}(0.25)$ |
| Meta-EGN-TACO (seed=4, step=1) | $1.01203_{\pm 0.01211}(0.13)$ | $1.00068_{\pm 0.00430}(0.11)$ |
| Meta-EGN-TACO (seed=8, step=3) | $1.00844_{\pm 0.01009}(0.50)$ | $1.00038_{\pm 0.003303}(0.43)$ |

Table 6: Mean ApR ($\uparrow$) and seconds per graph of COExpander and TACO-enhanced Meta-EGN on MC.

| Method | Twitter | COLLAB |
|---|---|---|
| COExpander (S=4, D=1, I=1)+Beam-16 | $0.96221_{\pm 0.06582}(0.02)$ | $0.97987_{\pm 0.07259}(0.01)$ |
| COExpander (S=4, D=20, I=20)+Beam-16 | $0.83018_{\pm 0.27615}(0.55)$ | $0.87452_{\pm 0.26720}(1.03)$ |
| COExpander (S=4, D=20, I=1) | $0.93072_{\pm 0.15479}(0.04)$ | $0.95374_{\pm 0.16264}(0.05)$ |
| Meta-EGN-TACO (seed=4, step=1) | $0.95913_{\pm 0.08458}(0.06)$ | $0.99764_{\pm 0.02393}(0.08)$ |
| Meta-EGN-TACO (seed=8, step=3) | $0.97197_{\pm 0.06979}(0.24)$ | $0.99916_{\pm 0.01116}(0.31)$ |

EGN-TACO achieves solutions that are slightly inferior yet broadly comparable to those of COExpander. On the MC task, however, Meta-EGN-TACO consistently outperforms COExpander across both datasets. A key advantage of TACO lies in its scalability with computation. While COExpander does not consistently benefit from increased runtime, TACO exhibits monotonic improvement as additional compute is allocated.

**Additional backbone.** To further validate the effectiveness of TACO, we also adopt ConsFormer (Xu et al., 2025) as the backbone and evaluate on the MaxCut problem. As ConsFormer is trained with a self-supervised objective, its setting closely aligns with ours. At test time, we fine-tune the trained model for 100 steps using the same objective. In terms of efficiency, ConsFormer runs for approximately 180 seconds for each graph (same setting as Xu et al. (2025)), while the average tuning time is only 0.15, 0.18, 0.37, and 2.87 seconds for graphs with $|V| = 800$, $|V| = 1K$, $|V| = 2K$, and $|V| \geq 3K$, respectively. Table 7 summarizes the results, where baseline performance is taken from Xu et al. (2025). We observe that standard fine-tuning provides no improvement over the trained model. In contrast, TACO consistently reduces the optimality gap across graphs of varying scales, demonstrating its effectiveness even on transformer-based architectures.

Table 7: Performance comparison for MaxCut on GSET (Ye, 2003). The numbers are the average percentage gap ($\downarrow$) to the best known cut size. "FT" stands for fine-tuning.

| Method | $|V| = 800$ | $|V| = 1K$ | $|V| = 2K$ | $|V| \geq 3K$ |
|---|---|---|---|---|
| Greedy | 5.26 | 6.64 | 6.81 | 6.30 |
| SDP | 3.14 | 4.24 | - | - |
| RUNCSP (Toenshoff et al., 2021) | 2.38 | 2.90 | 3.30 | 3.26 |
| ECO-DQN (Barrett et al., 2020) | 0.83 | 1.01 | 1.45 | 3.49 |
| ESCORD (Barrett et al., 2022) | 0.11 | 0.16 | 0.36 | 1.53 |
| ANYCSP (Tönshoff et al., 2023) | 0.02 | 0.05 | 0.12 | 0.42 |
| ConsFormer (reported) | 0.31 | 0.34 | 0.43 | 1.27 |
| ConsFormer (authors' code[1]) | 0.91 | 1.09 | 1.20 | 1.97 |
| ConsFormer-FT | 0.98 | 1.15 | 1.25 | 2.14 |
| ConsFormer-TACO | 0.80 | 0.58 | 0.70 | 1.59 |
| OR-Tools (Perron & Didier, 2025) | 1.84 | 2.09 | 3.38 | 3.08 |

**Distribution shift.** The detailed results are presented in Table 8. All models were tuned with 30 update steps, and 8 seeds were used. Models enhanced with TACO consistently demonstrate greater robustness to shift, maintaining better ApR than the fine-tuned counterparts. The EGN models without any additional optimization can only achieve 1.05976 on MVC and 0.90586 on MC, whereas the EGN models trained on RB200 and tested on RB200 achieve 1.02940 on MVC and 0.91045 on MC. For Meta-EGN, a similar

---

[1]https://github.com/khalil-research/ConsFormer.

Table 8: Mean ApR and seconds per graph of all methods under distribution shift.

| Method | MVC ($\downarrow$) | MC ($\uparrow$) |
|---|---|---|
| EGN | $1.05976_{\pm 0.00737}(0.18)$ | $0.90586_{\pm 0.11876}(0.06)$ |
| EGN-FT | $1.05453_{\pm 0.00692}(5.50)$ | $0.98558_{\pm 0.03514}(1.92)$ |
| EGN-FT-Online | $1.02505_{\pm 0.01135}(5.53)$ | $0.98148_{\pm 0.06051}(1.91)$ |
| EGN-TACO | $1.03659_{\pm 0.00617}(5.50)$ | $\mathbf{0.99166_{\pm 0.03770}(1.92)}$ |
| EGN-TACO-Online | $\mathbf{1.01958_{\pm 0.00454}(5.56)}$ | $0.98703_{\pm 0.04886}(1.95)$ |
| Meta-EGN | $1.04744_{\pm 0.00702}(0.18)$ | $0.90362_{\pm 0.10715}(0.06)$ |
| Meta-EGN-FT | $1.03044_{\pm 0.00591}(5.52)$ | $0.93951_{\pm 0.09678}(1.97)$ |
| Meta-EGN-FT-Online | $1.02354_{\pm 0.00553}(5.55)$ | $0.97317_{\pm 0.05384}(1.94)$ |
| Meta-EGN-TACO | $1.02875_{\pm 0.00533}(5.52)$ | $0.94491_{\pm 0.09504}(1.97)$ |
| Meta-EGN-TACO-Online | $\mathbf{1.01975_{\pm 0.00558}(5.56)}$ | $\mathbf{0.97328_{\pm 0.05842}(1.97)}$ |

Table 9: Mean ApR and seconds per graph of all methods on dynamic problems.

| Method | MVC ($\downarrow$) | MC ($\uparrow$) |
|---|---|---|
| EGN | $1.04315_{\pm 0.06230}(0.14)$ | $0.82964_{\pm 0.09868}(0.05)$ |
| EGN-FT | $1.01515_{\pm 0.04569}(4.33)$ | $0.95712_{\pm 0.08046}(1.42)$ |
| EGN-FT-Online | $1.01158_{\pm 0.03530}(4.35)$ | $1.00000_{\pm 0.00000}(1.38)$ |
| EGN-TACO | $1.01050_{\pm 0.03281}(4.33)$ | $0.98402_{\pm 0.05179}(1.42)$ |
| EGN-TACO-Online | $\mathbf{1.00852_{\pm 0.02950}(4.30)}$ | $\mathbf{1.00000_{\pm 0.00000}(1.38)}$ |
| Meta-EGN | $1.01244_{\pm 0.03563}(0.14)$ | $0.82533_{\pm 0.10378}(0.05)$ |
| Meta-EGN-FT | $0.99961_{\pm 0.01819}(4.36)$ | $0.98476_{\pm 0.05198}(1.59)$ |
| Meta-EGN-FT-Online | $1.01961_{\pm 0.04904}(4.34)$ | $0.99353_{\pm 0.03314}(1.46)$ |
| Meta-EGN-TACO | $\mathbf{0.99639_{\pm 0.01366}(4.36)}$ | $0.99015_{\pm 0.04546}(1.59)$ |
| Meta-EGN-TACO-Online | $1.00947_{\pm 0.03128}(4.34)$ | $\mathbf{0.99413_{\pm 0.03087}(1.38)}$ |

performance drop on MVC can be observed (1.04744 vs. 1.02502), but it remains robust on MC with distribution shift (0.90362 vs. 0.89779), which aligns with the findings of Wang & Li (2023).

**Dynamic problems.** As detailed in Table 9, models enhanced with TACO achieve superior performance on both the dynamic MVC and dynamic MC problems, maintaining the best mean ApRs. These results highlight TACO's effectiveness in guiding models toward high-quality solutions in evolving environments, thereby broadening its applicability to dynamic problem settings. Ideally, the online version is expected to work better than the standard version for dynamic problems, but this largely depends on the degree of problem-specific structural change in the graph snapshots over time: when there is little structural overlap, the parameters from the previous snapshot would be less useful (Liao et al., 2025).

**Sensitivity analysis on SP parameters.** To validate the generality and robustness of TACO, we selected the SP parameters relatively uniformly across datasets with limited tuning. This ensures that the observed performance gains are not the result of dataset-specific overfitting, but instead stem from the effectiveness of TACO. We include results of TACO with different sets of SP parameters in Table 15 in Appendix B.3. The results demonstrate that TACO consistently outperforms baselines across different parameter choices, suggesting that TACO is not overly sensitive to hyperparameter settings. This robustness reduces the burden of careful hyperparameter tuning in practical deployment. In Appendix B.6, we also explore adding perturbation during gradient updates at test time (using a variant of PGD (Jin et al., 2017)) as a possible alternative for introducing stochasticity, and show that TACO consistently yields superior solutions than PGD; further, we find that combining TACO and PGD achieves additional improvements over PGD alone.

# 5 Related work

The supervised learning paradigm has been shown to be powerful in NCO (Joshi et al., 2019; 2022; Vinyals et al., 2015; Gasse et al., 2019; Sun & Yang, 2023; Hudson et al., 2022; Li et al., 2023; 2024; Ma et al., 2025). These methods train models to predict high-quality solutions by leveraging large datasets of problem

instances annotated with optimal or near-optimal solutions. However, producing such labels is computationally expensive, particularly for large-scale instances.

Unsupervised learning (UL) and reinforcement learning (RL) approaches have been proposed to mitigate this dependency on labeled data (Bello et al., 2017; Khalil et al., 2017; Kool et al., 2019; Karalias & Loukas, 2020; Qiu et al., 2022; Toenshoff et al., 2021; Tönshoff et al., 2023; Wang & Li, 2023; Sanokowski et al., 2023). Despite this advantage, most UL and RL-based approaches still rely on extensive offline training across large datasets to learn *heuristics that generalize* across instances. An alternative instance-specific paradigm was introduced by Schuetz et al. (2022), who proposed an unsupervised framework that learns *instance-specific heuristics* by directly optimizing the combinatorial objective on a per-instance basis. This approach bypasses the need for offline training entirely, enabling the model to adapt to individual problem instances at test time. Follow-up works have enhanced this framework by improving solution quality, incorporating higher-order reasoning, and addressing dynamic CO problems (Heydaribeni et al., 2024; Ichikawa, 2024; Liao et al., 2025), achieving robust performance even on large-scale graphs.

Our approach is also related to Test-Time Training (Sun et al., 2020), which enhances supervised models during inference by optimizing on an auxiliary self-supervised task. However, in UL-based NCO, where models are trained using an unsupervised problem-specific objective, an auxiliary task is not needed. Instead, the UL objective used in training can be reused to guide test-time adaptation. More broadly, our method falls under the umbrella of the Test-Time Adaptation paradigm (Liang et al., 2025), which seeks to adapt trained models at test-time. In the NCO domain, prior works have primarily focused on improving solution quality during inference for RL-based approaches. Hottung et al. (2022) developed Efficient Active Search that updates a subset of model parameters for each test instance. Meta-SAGE (Son et al., 2023) adapts the model at test-time for better scalability. COMPASS (Chalumeau et al., 2023) employs search in a latent space to enable instance-specific policy adaptation. Our work extends the frontier of test-time adaptation to unsupervised NCO. In contrast to Meta-EGN, we accomplish effective adaptation through the lens of principled warm-starting and simultaneously unify generalizable and instance-specific NCO.

## 6   Discussion

**Conclusion.** We identified a fundamental incompatibility between generalization-focused unsupervised NCO models and instance-wise optimization and introduced TACO, a model-agnostic test-time adaptation framework that bridges the gap between the two paradigms. By viewing instance-wise adaptation from a warm-starting perspective, TACO combines the strengths of both paradigms, leveraging learned hypotheses while enabling effective instance-level refinement. Our extensive experiments on classical problems, minimum vertex cover, maximum clique, maximum independent set, and max cut, demonstrate that TACO consistently improves solution quality across static, distribution-shifted, and dynamic settings, all while incurring negligible computational overhead compared to standard fine-tuning. These results highlight the broad applicability and practical benefits of integrating TACO into unsupervised NCO pipelines.

**Limitations and future work.** The reported runtimes could be substantially reduced by enhancing the backbones, EGN and Meta-EGN, through parallelization of the seed dimension, adoption of more sophisticated input feature designs, and more efficient decoding mechanisms. Additionally, if batch data is available at test time, curriculum learning (Bengio et al., 2009; Lisicki et al., 2020; Liu et al., 2024) could be incorporated into TACO. More advanced instance-specific optimization methods extending PI-GNN (Ichikawa, 2024; Ichikawa & Iwashita, 2025) may further improve solution quality. Exploring training strategies that explicitly promote compatibility with TACO could also potentially accelerate convergence and enable fast transfer across related CO problems.

### Acknowledgments

This work is supported in part by the UCSD ECE Mentored Early Career Multiplier Award. We are grateful to the Action Editor and anonymous reviewers for their thoughtful comments and feedback.

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

# A  Additional Implementation details

We provide the exact loss function used in our experiments. For detailed derivations, please refer to Karalias & Loukas (2020) and Wang & Li (2023). The MVC loss is defined as:

$$\ell_{\mathrm{MVC}}(D; G) = \sum_{i=1}^{n} x_i + \beta \sum_{(i,j) \in E} (1 - x_i)(1 - x_j).$$

We adopted the simplified MC loss same as the implementation by Karalias & Loukas (2020):

$$\ell_{\mathrm{MC}}(D; G) = -\frac{1}{2} \sum_{(i,j) \in E} x_i x_j + \frac{\beta}{2} \sum_{i \neq j} x_i x_j.$$

We followed the training settings described in Wang & Li (2023). $\beta$ was set to 0.5 for the MVC experiments and 4 for the MC experiments. For tuning the trained models, we set $\beta = 0.5$ and $\lambda_{\mathrm{perturb}} = 0.001$ in all experiments; we used 0.0001 as the learning rate for tuning EGN models for MVC, 0.001 for MC, and 0.001 for Meta-EGN models for both problems.

**Hyperparameters for static problems.** The shrink parameter used in all experiments is listed in Tables 10 and 11.

Table 10: $\lambda_{\mathrm{shrink}}$ used in experiments for static MVC problems.

| Method | Twitter | | COLLAB | | RB200 | | RB500 | |
|---|---|---|---|---|---|---|---|---|
| | $\lambda_{\mathrm{shrink}}$ | $\lambda_{\mathrm{shrink\text{-}online}}$ | $\lambda_{\mathrm{shrink}}$ | $\lambda_{\mathrm{shrink\text{-}online}}$ | $\lambda_{\mathrm{shrink}}$ | $\lambda_{\mathrm{shrink\text{-}online}}$ | $\lambda_{\mathrm{shrink}}$ | $\lambda_{\mathrm{shrink\text{-}online}}$ |
| EGN-TACO | 0.3 | - | 0.3 | - | 0.3 | - | 0.5 | - |
| EGN-TACO-Online | 0.3 | 0.99 | 0.3 | 0.99 | 0.3 | 0.99 | 0.5 | 0.99 |
| Meta-EGN-TACO | 0.7 | - | 0.7 | - | 0.7 | - | 0.9 | - |
| Meta-EGN-TACO-Online | 0.7 | 0.9 | 0.7 | 0.9 | 0.7 | 0.9 | 0.9 | 0.9 |

Table 11: $\lambda_{\mathrm{shrink}}$ used in experiments for static MC problems.

| Method | Twitter | | COLLAB | | RB200 | | RB500 | |
|---|---|---|---|---|---|---|---|---|
| | $\lambda_{\mathrm{shrink}}$ | $\lambda_{\mathrm{shrink\text{-}online}}$ | $\lambda_{\mathrm{shrink}}$ | $\lambda_{\mathrm{shrink\text{-}online}}$ | $\lambda_{\mathrm{shrink}}$ | $\lambda_{\mathrm{shrink\text{-}online}}$ | $\lambda_{\mathrm{shrink}}$ | $\lambda_{\mathrm{shrink\text{-}online}}$ |
| EGN-TACO | 0.3 | - | 0.3 | - | 0.3 | - | 0.5 | - |
| EGN-TACO-Online | 0.3 | 0.99 | 0.3 | 0.99 | 0.3 | 0.99 | 0.5 | 0.99 |
| Meta-EGN-TACO | 0.7 | - | 0.7 | - | 0.7 | - | 0.7 | - |
| Meta-EGN-TACO-Online | 0.7 | 0.9 | 0.7 | 0.9 | 0.7 | 0.9 | 0.7 | 0.99 |

**Hyperparameters for problems with distribution shift.** The shrink parameter used in all experiments is listed in Tables 12.

Table 12: $\lambda_{\mathrm{shrink}}$ used in experiments for distribution shift.

| Method | MVC | | MC | |
|---|---|---|---|---|
| | $\lambda_{\mathrm{shrink}}$ | $\lambda_{\mathrm{shrink\text{-}online}}$ | $\lambda_{\mathrm{shrink}}$ | $\lambda_{\mathrm{shrink\text{-}online}}$ |
| EGN-TACO | 0.3 | - | 0.3 | - |
| EGN-TACO-Online | 0.3 | 0.99 | 0.3 | 0.99 |
| Meta-EGN-TACO | 0.7 | - | 0.7 | - |
| Meta-EGN-TACO-Online | 0.7 | 0.9 | 0.7 | 0.9 |

**Hyperparameters for dynamic problems.** The shrink parameter used in all experiments is listed in Tables 13.

Table 13: $\lambda_{\text{shrink}}$ used in experiments for dynamic problems.

| | MVC | | MC | |
|---|---|---|---|---|
| Method | $\lambda_{\text{shrink}}$ | $\lambda_{\text{shrink-online}}$ | $\lambda_{\text{shrink}}$ | $\lambda_{\text{shrink-online}}$ |
| EGN-TACO | 0.5 | - | 0.5 | - |
| EGN-TACO-Online | 0.5 | 1 | 0.5 | 1 |
| Meta-EGN-TACO | 0.5 | - | 0.5 | - |
| Meta-EGN-TACO-Online | 0.5 | 1 | 0.5 | 1 |

All models were implemented using PyTorch (Paszke et al., 2019) and PyTorch Geometric (Fey & Lenssen, 2019). Experiments were conducted on a machine with a single NVIDIA GeForce RTX 4090 GPU, a 32-core Intel Core i9-14900K CPU, and 64 GB of RAM running Ubuntu 24.04.

# B    Additional results

## B.1    Full-range plot showing EGN optimization trajectories

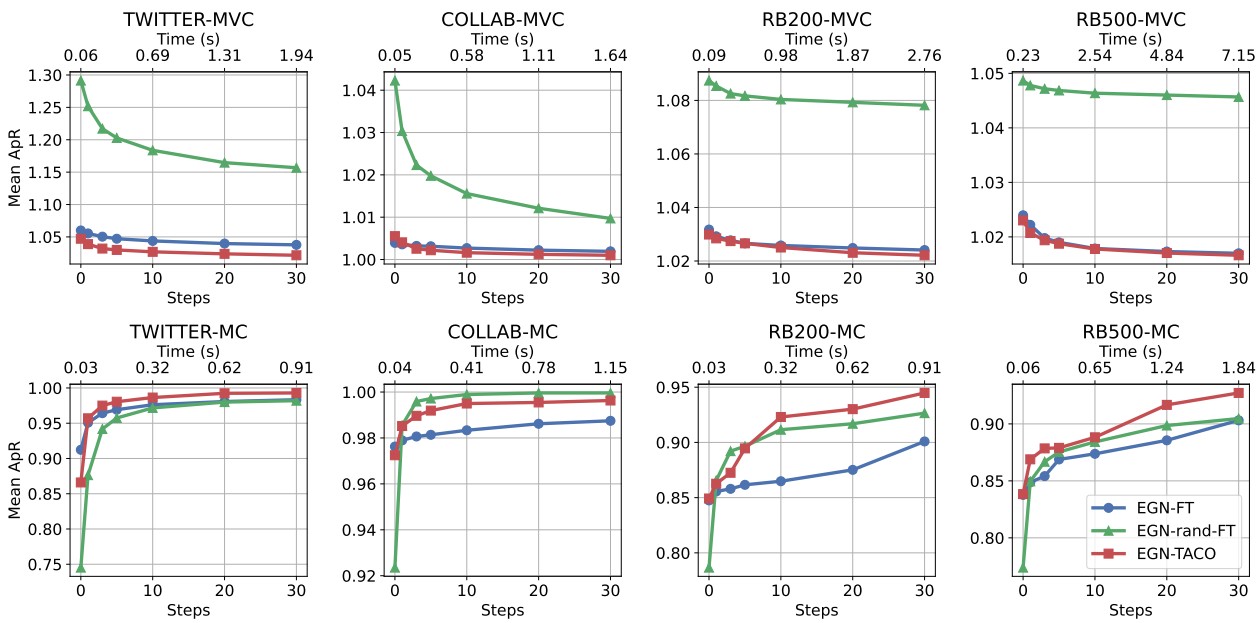

Figure 5: Mean ApR of methods using EGN as the backbone on static MVC ($\downarrow$) and MC ($\uparrow$) problems with respect to the number of update steps. "FT" stands for fine-tuning; "rand" means models are freshly initialized. The wall clock time factors in the decoding operations.

## B.2 Comparison with EGN with 256 seeds

Table 14: Mean ApR and seconds per graph of EGN with 256 seeds and TACO-enhanced EGN with 8 seeds.

| TWITTER | | |
| --- | --- | --- |
| Method | MVC ($\downarrow$) | MC ($\uparrow$) |
| EGN (256) | $1.02875_{\pm 0.02073}(3.37)$ | $0.99167_{\pm 0.02903}(1.03)$ |
| EGN-TACO (8) | $1.01668_{\pm 0.01636}(3.90)$ | $0.99766_{\pm 0.01310}(1.86)$ |
| EGN-TACO-Online (8) | $1.01357_{\pm 0.01417}(3.96)$ | $0.99277_{\pm 0.03325}(1.90)$ |
| COLLAB | | |
| Method | MVC ($\downarrow$) | MC ($\uparrow$) |
| EGN (256) | $1.00040_{\pm 0.00319}(2.70)$ | $1.00000_{\pm 0.00000}(1.68)$ |
| EGN-TACO (8) | $1.00040_{\pm 0.00341}(3.32)$ | $0.99976_{\pm 0.00580}(2.36)$ |
| EGN-TACO-Online (8) | $1.00017_{\pm 0.00216}(3.33)$ | $1.00000_{\pm 0.00000}(2.40)$ |
| RB200 | | |
| Method | MVC ($\downarrow$) | MC ($\uparrow$) |
| EGN (256) | $1.02071_{\pm 0.00405}(5.01)$ | $0.99575_{\pm 0.01858}(1.07)$ |
| EGN-TACO (8) | $1.02052_{\pm 0.00420}(5.52)$ | $0.98094_{\pm 0.05496}(1.88)$ |
| EGN-TACO-Online (8) | $1.01930_{\pm 0.00461}(5.55)$ | $0.98465_{\pm 0.05296}(1.88)$ |
| RB500 | | |
| Method | MVC ($\downarrow$) | MC ($\uparrow$) |
| EGN (256) | $1.01932_{\pm 0.00177}(13.77)$ | $0.99635_{\pm 0.01497}(2.44)$ |
| EGN-TACO (8) | $1.01577_{\pm 0.00214}(14.24)$ | $0.97101_{\pm 0.07950}(3.75)$ |
| EGN-TACO-Online (8) | $1.01512_{\pm 0.00209}(14.25)$ | $0.98684_{\pm 0.04781}(3.88)$ |

## B.3 Sensitivity analysis on SP parameters

Table 15: Mean ApR of EGN-TACO with different sets of SP parameters. $\lambda_{\text{shrink}} = 1, \lambda_{\text{perturb}} = 0$ is equivalent to EGN-FT; $\lambda_{\text{shrink}} = 0, \lambda_{\text{perturb}} = 1$ is equivalent to EGN-rand-FT. All settings use 30 update steps and 8 random seeds.

| $\lambda_{\text{shrink}}$ | $\lambda_{\text{perturb}}$ | TWITTER-MVC ($\downarrow$) | TWITTER-MC ($\uparrow$) | COLLAB-MVC ($\downarrow$) | COLLAB-MC ($\uparrow$) |
| --- | --- | --- | --- | --- | --- |
| 0.0 | 0.001 | $1.05593_{\pm 0.06882}$ | $0.67702_{\pm 0.24641}$ | $1.01325_{\pm 0.03691}$ | $0.84874_{\pm 0.28305}$ |
| 0.1 | 0.001 | $1.01233_{\pm 0.01254}$ | $0.99678_{\pm 0.01445}$ | $1.00023_{\pm 0.00243}$ | $0.99983_{\pm 0.00527}$ |
| 0.3 | 0.001 | $1.01668_{\pm 0.01636}$ | $0.99766_{\pm 0.01310}$ | $1.00040_{\pm 0.00341}$ | $0.99976_{\pm 0.00580}$ |
| 0.5 | 0.001 | $1.02221_{\pm 0.01871}$ | $0.99507_{\pm 0.03153}$ | $1.00039_{\pm 0.00342}$ | $0.99954_{\pm 0.00900}$ |
| 0.7 | 0.001 | $1.02563_{\pm 0.01986}$ | $0.99256_{\pm 0.03468}$ | $1.00043_{\pm 0.00336}$ | $0.99954_{\pm 0.00900}$ |
| 0.9 | 0.001 | $1.02845_{\pm 0.02093}$ | $0.99193_{\pm 0.04797}$ | $1.00048_{\pm 0.00369}$ | $0.99916_{\pm 0.01238}$ |
| 0.3 | 0.0001 | $1.01690_{\pm 0.01646}$ | $0.99781_{\pm 0.01355}$ | $1.00039_{\pm 0.00327}$ | $0.99956_{\pm 0.00857}$ |
| 0.3 | 0.001 | $1.01668_{\pm 0.01636}$ | $0.99766_{\pm 0.01310}$ | $1.00040_{\pm 0.00341}$ | $0.99976_{\pm 0.00580}$ |
| 0.3 | 0.01 | $1.01646_{\pm 0.01568}$ | $0.99775_{\pm 0.01056}$ | $1.00032_{\pm 0.00299}$ | $0.99954_{\pm 0.00900}$ |
| 0.3 | 0.1 | $1.02063_{\pm 0.01765}$ | $0.99827_{\pm 0.01340}$ | $1.00028_{\pm 0.00265}$ | $0.99929_{\pm 0.01197}$ |
| 1.0 | 0.0 | $1.03026_{\pm 0.02123}$ | $0.99151_{\pm 0.04802}$ | $1.00071_{\pm 0.00442}$ | $0.99916_{\pm 0.01238}$ |
| 0.0 | 1.0 | $1.12846_{\pm 0.06283}$ | $0.99430_{\pm 0.02285}$ | $1.00611_{\pm 0.01722}$ | $1.00000_{\pm 0.00000}$ |

### B.4 EGN performance averaged over all seeds

Table 16: Mean ApR ($\downarrow$) of EGN and TACO-enhanced EGN with 8 seeds on MVC. All results are from 30 update steps. "(best)" means the best-performing seed is reported; "(avg)" means the performance is averaged over all seeds.

| Method | Twitter | COLLAB | RB200 | RB500 |
|---|---|---|---|---|
| EGN (best) | $1.03026_{\pm 0.02123}$ | $1.00071_{\pm 0.00442}$ | $1.02243_{\pm 0.00495}$ | $1.01630_{\pm 0.00208}$ |
| EGN-TACO (best) | $1.01668_{\pm 0.01636}$ | $1.00040_{\pm 0.00341}$ | $1.02052_{\pm 0.00420}$ | $1.01577_{\pm 0.00214}$ |
| EGN (avg) | $1.06519_{\pm 0.02900}$ | $1.00920_{\pm 0.02128}$ | $1.03466_{\pm 0.00359}$ | $1.02169_{\pm 0.00216}$ |
| EGN-TACO (avg) | $1.04388_{\pm 0.02885}$ | $1.00852_{\pm 0.02034}$ | $1.03385_{\pm 0.00329}$ | $1.02136_{\pm 0.00209}$ |

Table 17: Mean ApR ($\uparrow$) of EGN and TACO-enhanced EGN with 8 seeds on MC. All results are from 30 update steps. "(best)" means the best-performing seed is reported; "(avg)" means the performance is averaged over all seeds.

| Method | Twitter | COLLAB | RB200 | RB500 |
|---|---|---|---|---|
| EGN (best) | $0.99151_{\pm 0.04802}$ | $0.99916_{\pm 0.01238}$ | $0.95659_{\pm 0.09698}$ | $0.95306_{\pm 0.11117}$ |
| EGN-TACO (best) | $0.99766_{\pm 0.01310}$ | $0.99976_{\pm 0.00580}$ | $0.98094_{\pm 0.05496}$ | $0.97101_{\pm 0.07950}$ |
| EGN (avg) | $0.89725_{\pm 0.09001}$ | $0.90530_{\pm 0.12583}$ | $0.78436_{\pm 0.11440}$ | $0.81894_{\pm 0.17828}$ |
| EGN-TACO (avg) | $0.90888_{\pm 0.07366}$ | $0.95183_{\pm 0.08028}$ | $0.79625_{\pm 0.10480}$ | $0.81644_{\pm 0.17891}$ |

## B.5 Experiments on the maximum independent set problem

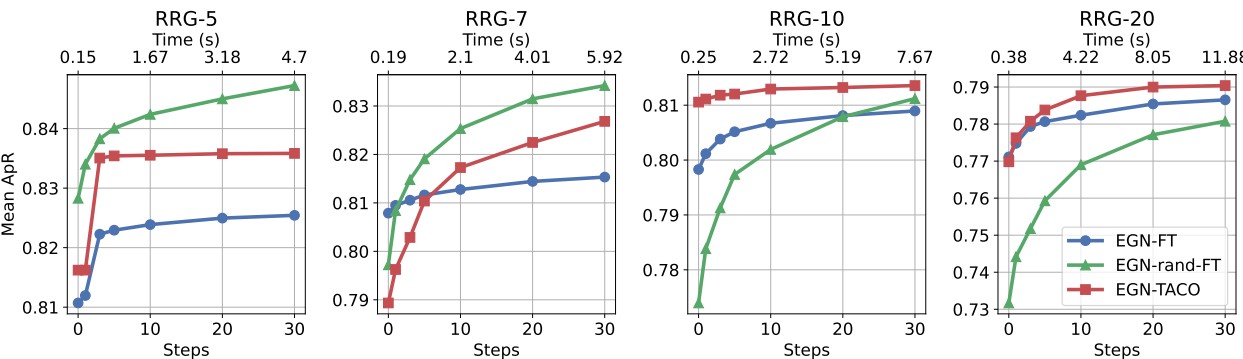

Figure 6: Mean ApR ($\uparrow$) of methods using EGN as the backbone on static MIS problems with respect to the number of update steps. "FT" stands for fine-tuning; "rand" means models are freshly initialized. The wall clock time factors in the decoding operations.

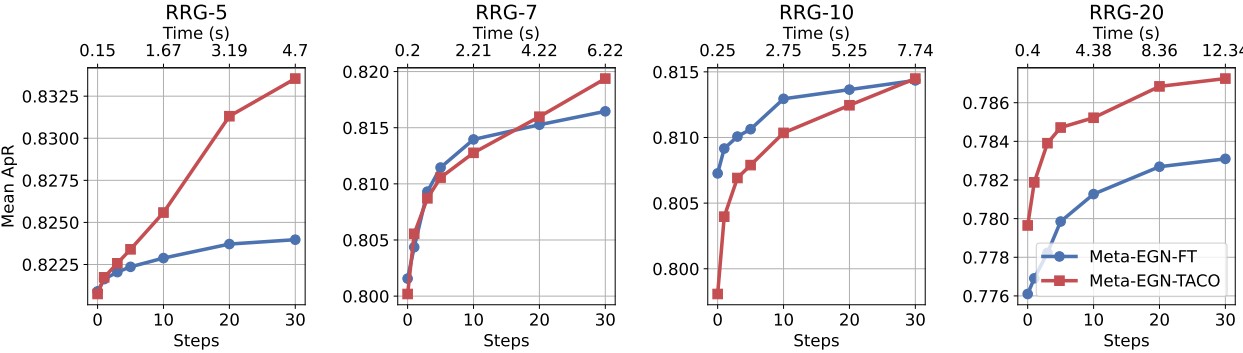

Figure 7: Mean ApR ($\uparrow$) of methods using Meta-EGN as the backbone on static MIS problems with respect to the number of update steps. "FT" stands for fine-tuning. The wall clock time factors in the decoding operations.

Following Wang & Li (2023), we generated random regular graphs (RRGs) with 1000 nodes for the maximum independent set (MIS) experiments. We constructed RRG-d datasets where $d \in \{5, 7, 10, 20\}$ denotes the degree of each node. As observed by Wang & Li (2023), models trained on RRG-20 can generalize better to RRGs of other degrees than models trained on lower-degree RRGs (e.g., degree 3). Given this phenomenon, we trained our models on RRG-20 and evaluated them on RRG-d where $d \in \{5, 7, 10, 20\}$.

For training, we generated 3000 RRG-20 instances; for evaluation, we generated 50 graphs for each RRG-d. Since RRGs have theoretical upper bounds on MIS size (Duckworth & Zito, 2009), we used them to compute the ApRs of each solution, same as the evaluation protocol of Schuetz et al. (2022) and Wang & Li (2023).

Consistent with Wang & Li (2023), the models were trained to optimize

$$\ell_{\mathrm{MIS}}(D; G) = -\sum_{i=1}^{n} x_i + \beta \sum_{(i,j) \in E} x_i x_j.$$

$\beta$ was set to 0.5. For tuning the trained models, we set $\beta = 0.5$, $\lambda_{shrink} = 0.3$, and $\lambda_{perturb} = 0.001$ in all experiments, except for RRG-10 where $\lambda_{shrink}$ was set to 0.5 when tuning the trained Meta-EGN model; we used 0.0001 as the learning rate for tuning all models.

Figures 6 and 7 illustrate the optimization trajectory of each model during test-time update. Table 18 reports the solution quality of each method after 30 update steps. Our MIS results show that TACO consistently

Table 18: Mean ApR ($\uparrow$) and seconds per graph of all methods on MIS. All results are from 30 update steps. "FT" stands for fine-tuning; "Accurate (8)" means 8 seeds were used. The best solutions are in **bold**, and the cases where EGN + TACO outperform Meta-EGN-FT are in gray.

| RRG-5 | | |
|---|---|---|
| Method | Fast (1) | Balanced (4) | Accurate (8) |
| EGN | $0.80025_{\pm 0.01315}(0.04)$ | $0.81071_{\pm 0.00904}(0.15)$ | $0.81659_{\pm 0.00743}(0.30)$ |
| EGN-FT | $0.81534_{\pm 0.01289}(1.18)$ | $0.82543_{\pm 0.00903}(4.70)$ | $0.83084_{\pm 0.00775}(9.40)$ |
| EGN-TACO | $\mathbf{0.82522}_{\pm \mathbf{0.01110}}(\mathbf{1.21})$ | $\mathbf{0.83583}_{\pm \mathbf{0.00944}}(\mathbf{4.85})$ | $\mathbf{0.83968}_{\pm \mathbf{0.00931}}(\mathbf{9.69})$ |
| Meta-EGN | $0.80961_{\pm 0.01340}(0.04)$ | $0.82090_{\pm 0.00952}(0.15)$ | $0.82642_{\pm 0.00844}(0.30)$ |
| Meta-EGN-FT | $0.81326_{\pm 0.01267}(1.18)$ | $0.82397_{\pm 0.00930}(4.70)$ | $0.82933_{\pm 0.00805}(9.41)$ |
| Meta-EGN-TACO | $\mathbf{0.82137}_{\pm \mathbf{0.01361}}(\mathbf{1.21})$ | $\mathbf{0.83355}_{\pm \mathbf{0.00765}}(\mathbf{4.85})$ | $\mathbf{0.83797}_{\pm \mathbf{0.00674}}(\mathbf{9.70})$ |

| RRG-7 | | |
|---|---|---|
| Method | Fast (1) | Balanced (4) | Accurate (8) |
| EGN | $0.78964_{\pm 0.01622}(0.05)$ | $0.80788_{\pm 0.01169}(0.19)$ | $0.81306_{\pm 0.00792}(0.38)$ |
| EGN-FT | $0.79852_{\pm 0.01467}(1.48)$ | $0.81532_{\pm 0.00933}(5.92)$ | $0.81884_{\pm 0.00849}(11.84)$ |
| EGN-TACO | $\mathbf{0.81783}_{\pm \mathbf{0.01058}}(\mathbf{1.53})$ | $\mathbf{0.82682}_{\pm \mathbf{0.00824}}(\mathbf{6.13})$ | $\mathbf{0.83141}_{\pm \mathbf{0.00630}}(\mathbf{12.26})$ |
| Meta-EGN | $0.79042_{\pm 0.01398}(0.05)$ | $0.80156_{\pm 0.00846}(0.20)$ | $0.80823_{\pm 0.00872}(0.40)$ |
| Meta-EGN-FT | $0.80782_{\pm 0.00898}(1.56)$ | $0.81646_{\pm 0.00691}(6.22)$ | $0.82158_{\pm 0.00648}(12.45)$ |
| Meta-EGN-TACO | $\mathbf{0.80996}_{\pm \mathbf{0.01024}}(\mathbf{1.55})$ | $\mathbf{0.81938}_{\pm \mathbf{0.00717}}(\mathbf{6.18})$ | $\mathbf{0.82384}_{\pm \mathbf{0.00661}}(\mathbf{12.36})$ |

| RRG-10 | | |
|---|---|---|
| Method | Fast (1) | Balanced (4) | Accurate (8) |
| EGN | $0.78125_{\pm 0.01657}(0.06)$ | $0.79829_{\pm 0.01168}(0.25)$ | $0.80551_{\pm 0.00875}(0.49)$ |
| EGN-FT | $0.79534_{\pm 0.01288}(1.92)$ | $0.80895_{\pm 0.00759}(7.67)$ | $0.81449_{\pm 0.00688}(15.34)$ |
| EGN-TACO | $\mathbf{0.80215}_{\pm \mathbf{0.01288}}(\mathbf{1.91})$ | $\mathbf{0.81358}_{\pm \mathbf{0.00812}}(\mathbf{7.64})$ | $\mathbf{0.81757}_{\pm \mathbf{0.00700}}(\mathbf{15.28})$ |
| Meta-EGN | $0.78525_{\pm 0.01517}(0.06)$ | $0.80726_{\pm 0.01085}(0.25)$ | $0.81231_{\pm 0.00980}(0.50)$ |
| Meta-EGN-FT | $\mathbf{0.79555}_{\pm \mathbf{0.01444}}(\mathbf{1.94})$ | $\mathbf{0.81435}_{\pm \mathbf{0.01013}}(\mathbf{7.74})$ | $\mathbf{0.81968}_{\pm \mathbf{0.00885}}(\mathbf{15.49})$ |
| Meta-EGN-TACO | $0.78931_{\pm 0.02287}(1.96)$ | $0.81196_{\pm 0.01146}(7.84)$ | $0.81912_{\pm 0.00927}(15.69)$ |

| RRG-20 | | |
|---|---|---|
| Method | Fast (1) | Balanced (4) | Accurate (8) |
| EGN | $0.74823_{\pm 0.01686}(0.10)$ | $0.77113_{\pm 0.01153}(0.38)$ | $0.77701_{\pm 0.01051}(0.77)$ |
| EGN-FT | $0.76931_{\pm 0.01379}(2.97)$ | $0.78654_{\pm 0.01044}(11.88)$ | $0.79252_{\pm 0.00956}(23.76)$ |
| EGN-TACO | $\mathbf{0.77235}_{\pm \mathbf{0.01570}}(\mathbf{2.96})$ | $\mathbf{0.79039}_{\pm \mathbf{0.01023}}(\mathbf{11.83})$ | $\mathbf{0.79708}_{\pm \mathbf{0.00967}}(\mathbf{23.67})$ |
| Meta-EGN | $0.75329_{\pm 0.01938}(0.10)$ | $0.77610_{\pm 0.01262}(0.40)$ | $0.78482_{\pm 0.01160}(0.80)$ |
| Meta-EGN-FT | $0.76363_{\pm 0.01751}(3.08)$ | $0.78309_{\pm 0.01230}(12.34)$ | $0.79110_{\pm 0.01059}(24.67)$ |
| Meta-EGN-TACO | $\mathbf{0.76890}_{\pm \mathbf{0.01564}}(\mathbf{3.03})$ | $\mathbf{0.78725}_{\pm \mathbf{0.01095}}(\mathbf{12.13})$ | $\mathbf{0.79374}_{\pm \mathbf{0.01146}}(\mathbf{24.27})$ |

outperforms naive fine-tuning and narrows or overcomes the performance gap between instance-specific baselines, mirroring the observations for the MVC and MC experiments. One exception is Meta-EGN on the RRG-10 dataset. We note that our models were only trained on RRGs of degree 20, and Meta-EGN-TACO is still rapidly improving and on the verge of surpassing the fine-tuning baseline at step 30, suggesting that additional update steps would likely close the remaining gap, as shown in Figure 7. Moreover, Table 18 highlights the effectiveness of TACO in test-time adaptation over MAML.

### B.6 Perturbed gradient descent

TACO applies a transformation to the trained model parameters once prior to adaptation. An alternative approach is to introduce perturbations during the test-time gradient update steps. Jin et al. (2017) theoretically analyze how perturbed gradient descent (PGD) facilitates escaping saddle points. Motivated by this, we implement a simplified PGD variant by injecting perturbations into the gradient updates at test time. As shown in Tables 19 and 20, PGD can occasionally reach better optima than standard fine-tuning. However, TACO (which introduces perturbations before gradient descent updates) consistently yields superior solutions than PGD (which introduces perturbation during gradient descent updates); further, we find that TACO and PGD can be combined to achieve additional improvements over PGD alone.

Table 19: Performance ($\downarrow$) comparison between TACO and PGD on MVC on Twitter. "FT" stands for fine-tuning. All results are from 30 update steps.

| Method | Fast (1) | Balanced (4) | Accurate (8) |
|---|---|---|---|
| EGN | $1.09349_{\pm 0.05062}(0.02)$ | $1.05996_{\pm 0.03552}(0.06)$ | $1.04928_{\pm 0.02902}(0.13)$ |
| EGN-PGD | $1.05578_{\pm 0.03643}(0.53)$ | $1.03181_{\pm 0.02226}(2.11)$ | $1.02561_{\pm 0.02014}(4.21)$ |
| EGN-TACO-PGD | $1.04290_{\pm 0.03599}(0.53)$ | $1.02500_{\pm 0.02239}(2.13)$ | $1.01894_{\pm 0.01682}(4.25)$ |
| EGN-FT | $1.06311_{\pm 0.03905}(0.48)$ | $1.03775_{\pm 0.02640}(1.94)$ | $1.03026_{\pm 0.02123}(3.88)$ |
| EGN-TACO | $\mathbf{1.03826_{\pm 0.03473}(0.49)}$ | $\mathbf{1.02165_{\pm 0.02221}(1.95)}$ | $\mathbf{1.01668_{\pm 0.01636}(3.90)}$ |

Table 20: Performance ($\uparrow$) comparison between TACO and PGD on MC on Twitter. "FT" stands for fine-tuning. All results are from 30 update steps.

| Method | Fast (1) | Balanced (4) | Accurate (8) |
|---|---|---|---|
| EGN | $0.73856_{\pm 0.25477}(0.01)$ | $0.91233_{\pm 0.12556}(0.03)$ | $0.95073_{\pm 0.08131}(0.06)$ |
| EGN-PGD | $0.91600_{\pm 0.15072}(0.26)$ | $0.97762_{\pm 0.06759}(1.05)$ | $0.98938_{\pm 0.04967}(2.10)$ |
| EGN-TACO-PGD | $0.91683_{\pm 0.14757}(0.26)$ | $0.98507_{\pm 0.04729}(1.04)$ | $0.99257_{\pm 0.03345}(2.08)$ |
| EGN-FT | $0.93744_{\pm 0.13083}(0.23)$ | $0.98332_{\pm 0.06204}(0.91)$ | $0.99151_{\pm 0.04802}(1.83)$ |
| EGN-TACO | $\mathbf{0.95419_{\pm 0.10578}(0.23)}$ | $\mathbf{0.99295_{\pm 0.02656}(0.93)}$ | $\mathbf{0.99766_{\pm 0.01310}(1.86)}$ |

