# OpenReview forum: "Test-Time Adaptation for Unsupervised Combinatorial Optimization"
_TMLR — Accepted by TMLR_

### Review · Reviewer_r3t2 · 2026-01-29

**Summary Of Contributions:**

The paper proposes a shrink-and-perturb (SP) fine-tuning method for machine learning models that are trained to generate solutions to combinatorial optimization problems. Following unsupervised (generative) model training, we are presented with an instance of the combinatorial problem or a sequence of instances. Because the instance is likely new to the model, it would make sense that some test-time adaptation is performed in order to optimize solution quality. Naive fine-tuning takes gradient descent steps on the same loss function used for training, only applied to the test instance at hand. SP, on the other hand, "shrinks" the model's parameters by multiplying them with a scalar between 0 and 1, then perturbs them by adding random noise. Following SP, gradient steps are taken from the perturbed initialization. SP is also better in principle than a method that operates strictly on the test instance without any training on potentially similar instances.

Experiments on minimium vertex cover and max clique show that SP outperforms naive fine-tuning or instance-specific learning across two different backbone architectures, on test instances from the training distribution, outside of it, and in an online setting.

The contributions are:
- The study of a practically relevant problem, namely test-time adaptation of trained models for combinatorial optimization;
- A simple adaptation method is proposed;
- A series of experiments across different graphs, combinatorial problems, backbone architectures, baselines, showing that the proposed SP method performs favorably.

**Audience:**

Yes

**Audience Explanation:**

Learning-to-optimize, particularly in the discrete or combinatorial setting, is a popular topic. TMLR has published in this area in the last few years, and thus there will be interest in the findings of this paper.

**Broader Impact Concerns:**

None to report.

**Claims And Evidence:**

Yes

**Claims Explanation:**

The experimental design is described in sufficient detail and is satisfactory. The experimental results do support the main claim of the paper, which is that the shrink-and-perturb heuristic for initializing a fine-tuning of a model trained for combinatorial optimization outperforms naive fine-tuning and instance-specific learning. This hypothesis seems to hold for the two problems and backbones herein. Little can be said about other unsupervised combinatorial optimization settings.

**Requested Changes:**

Major changes:

- Paper length: I believe the paper can be shortened to 8-9 pages. There is a lot of repetition of the main claims between the abstract, introduction, preliminaries, and method sections. The authors should aim to be concise given that the message of the paper is very clear and can be described compactly.

- Additional experiments: With the shortening I propose, you have more space to add another problem and/or backbone, which I believe is important to validating your hypothesis. I recommend looking at https://arxiv.org/abs/1909.10461 (code: https://github.com/RAISELab-atUVA/dnn-opf) and https://arxiv.org/abs/2502.15794 (code: https://github.com/khalil-research/ConsFormer) as very different examples of additional settings that your method should apply to. The results in the paper suggest that these MVC/MC instances are quite easy, as the optimality gap seems to be often close to 1% for most methods. This makes the contribution limited, unless further evidence is produced on additional problems.

Minor comments:

- p1: "diverse graph structures": why graphs? up to this point, you are talking about combinatorial problems in general, some of which are not typically defined on graphs.
- p2: "constraint-valid" --> constraint-feasible
- p4: "SP parameters". SP no defined at this point
- p4-5: Fig 1 is mentioned in page 4 before the "Mean ApR" metric on the vertical axis of the figure is described anywhere. Even the caption of the figure does not define it.
- p6: "setting the number of seeds to 4"; what does this mean? Are the reported results averaged over 4 seeds, or is this is a best-of strategy that reports the best solution found across 4 runs?

---

> ### Author Response · Authors · 2026-02-12
>
> We thank the reviewer for their thoughtful feedback. The manuscript has been revised accordingly, with all modifications highlighted in blue.
>
> **Addition problem**
>
> We thank the reviewer for the suggestion to validate TACO on additional problems/backbones. Our focus is within unsupervised NCO pipelines, with an emphasis on graph problems, rather than to exhaustively benchmark across all CO domains. In response, we have added experiments on the Maximum Independent Set (MIS) problem. We believe this approach provides a clearer and more principled validation of TACO’s generality within unsupervised NCO, while leaving extension to other domains and architectures as promising future work.
>
> In Appendix B.5, we added the optimization trajectories of different methods (Figures 6 and 7) and the solution quality of them after 30 update steps (Table 15). The table is also included below in this comment. All experiment details are provided in the updated manuscript along with the results. Our MIS results show that TACO consistently improves solution quality over naive fine-tuning and instance-specific baselines across datasets and update budgets, mirroring the optimization behavior observed for MVC and MC.
>
> RRG-5:
> | Method | Fast (1) | Balanced (4) | Accurate (8) |
> | --- | --- | --- | --- |
> EGN        | $0.80025_{\pm 0.01315} (0.04)$ | $0.81071_{\pm 0.00904} (0.15)$ | $0.81659_{\pm 0.00743} (0.30)$ |
> EGN-FT     | $0.81534_{\pm 0.01289} (1.18)$ | $0.82543_{\pm 0.00903} (4.70)$ | $0.83084_{\pm 0.00775} (9.40)$ |
> EGN-TACO   | $\mathbf{0.82522_{\pm 0.01110} (1.21)}$ | $\mathbf{0.83583_{\pm 0.00944} (4.85)}$ | $\mathbf{0.83968_{\pm 0.00931} (9.69)}$ |
> Meta-EGN      | $0.80961_{\pm 0.01340} (0.04)$ | $0.82090_{\pm 0.00952} (0.15)$ | $0.82642_{\pm 0.00844} (0.30)$ |
> Meta-EGN-FT   | $0.81326_{\pm 0.01267} (1.18)$ | $0.82397_{\pm 0.00930} (4.70)$ | $0.82933_{\pm 0.00805} (9.41)$ |
> Meta-EGN-TACO | $\mathbf{0.82137_{\pm 0.01361} (1.21)}$ | $\mathbf{0.83355_{\pm 0.00765} (4.85)}$ | $\mathbf{0.83797_{\pm 0.00674} (9.70)}$ |
>
> RRG-7
> | Method | Fast (1) | Balanced (4) | Accurate (8) |
> | --- | --- | --- | --- |
> EGN  | $0.78964_{\pm 0.01622} (0.05)$ | $0.80788_{\pm 0.01169} (0.19)$ | $0.81306_{\pm 0.00792} (0.38)$ |
> EGN-FT  | $0.79852_{\pm 0.01467} (1.48)$ | $0.81532_{\pm 0.00933} (5.92)$ | $0.81884_{\pm 0.00849} (11.84)$ |
> EGN-TACO  | $\mathbf{0.81783_{\pm 0.01058} (1.53)}$ | $\mathbf{0.82682_{\pm 0.00824} (6.13)}$ | $\mathbf{0.83141_{\pm 0.00630} (12.26)}$ |
> Meta-EGN | $0.79042_{\pm 0.01398} (0.05)$ | $0.80156_{\pm 0.00846} (0.20)$ | $0.80823_{\pm 0.00872} (0.40)$ |
> Meta-EGN-FT   | $0.80782_{\pm 0.00898} (1.56)$ | $0.81646_{\pm 0.00691} (6.22)$ | $0.82158_{\pm 0.00648} (12.45)$ |
> Meta-EGN-TACO | $\mathbf{0.80996_{\pm 0.01024} (1.55)}$ | $\mathbf{0.81938_{\pm 0.00717} (6.18)}$ | $\mathbf{0.82384_{\pm 0.00661} (12.36)}$ |
>
> RRG-10
> | Method | Fast (1) | Balanced (4) | Accurate (8) |
> | --- | --- | --- | --- |
> EGN  | $0.78125_{\pm 0.01657} (0.06)$ | $0.79829_{\pm 0.01168} (0.25)$ | $0.80551_{\pm 0.00875} (0.49)$ |
> EGN-FT  | $0.79534_{\pm 0.01288} (1.92)$ | $0.80895_{\pm 0.00759} (7.67)$ | $0.81449_{\pm 0.00688} (15.34)$ |
> EGN-TACO  | $\mathbf{0.80215_{\pm 0.01288} (1.91)}$ | $\mathbf{0.81358_{\pm 0.00812} (7.64)}$ | $\mathbf{0.81757_{\pm 0.00700} (15.28)}$ |
> Meta-EGN | $0.78525_{\pm 0.01517} (0.06)$ | $0.80726_{\pm 0.01085} (0.25)$ | $0.81231_{\pm 0.00980} (0.50)$ |
> Meta-EGN-FT | $\mathbf{0.79555_{\pm 0.01444} (1.94)}$ | $\mathbf{0.81435_{\pm 0.01013} (7.74)}$ | $\mathbf{0.81968_{\pm 0.00885} (15.49)}$ |
> Meta-EGN-TACO | $0.78931_{\pm 0.02287} (1.96)$ | $0.81196_{\pm 0.01146} (7.84)$ | $0.81912_{\pm 0.00927} (15.69)$ |
>
> RRG-20
> | Method | Fast (1) | Balanced (4) | Accurate (8) |
> | --- | --- | --- | --- |
> EGN | $0.74823_{\pm 0.01686} (0.10)$ | $0.77113_{\pm 0.01153} (0.38)$ | $0.77701_{\pm 0.01051} (0.77)$ |
> EGN-FT | $0.76931_{\pm 0.01379} (2.97)$ | $0.78654_{\pm 0.01044} (11.88)$ | $0.79252_{\pm 0.00956} (23.76)$ |
> EGN-TACO | $\mathbf{0.77235_{\pm 0.01570} (2.96)}$} | $\mathbf{0.79039_{\pm 0.01023} (11.83)}$} | $\mathbf{0.79708_{\pm 0.00967} (23.67)}$} |
> Meta-EGN  | $0.75329_{\pm 0.01938} (0.10)$ | $0.77610_{\pm 0.01262} (0.40)$ | $0.78482_{\pm 0.01160} (0.80)$ |
> Meta-EGN-FT | $0.76363_{\pm 0.01751} (3.08)$ | $0.78309_{\pm 0.01230} (12.34)$ | $0.79110_{\pm 0.01059} (24.67)$ |
> Meta-EGN-TACO | $\mathbf{0.76890_{\pm 0.01564} (3.03)}$ | $\mathbf{0.78725_{\pm 0.01095} (12.13)}$ | $\mathbf{0.79374_{\pm 0.01146} (24.27)}$ |
>
> **Minor comments**
> - We have changed "diverse graph structures" to "diverse problem structures".
> - Mean ApR is defined in the experiments section. To avoid confusion and repetition, we use "performance ($\uparrow$)" in the caption of Figure 1.
> - Our evaluation protocol follows established practice in prior unsupervised NCO work (e.g., Karalias & Loukas, 2020; Wang & Li, 2023), where multiple seeds are used, and the best-performing seed is reported. We also included the performance averaged over the random seeds in Appendix B4.

---

### Review · Reviewer_Ps7m · 2026-02-03

**Summary Of Contributions:**

This paper contributes to area of unsupervised neural combinatorial optimization. The contributions of this paper can be summarized as follows:

- The authors reveal a counterintuitive "incompatibility" where pre-trained unsupervised NCO models often underperform randomly initialized ones when naively fine-tuned at test time.

- (Main contribution) TACO Framework: To resolve this, they introduce TACO, a model-agnostic test-time adaptation framework that employs a "shrink and perturb" initialization strategy to effectively unify learned inductive biases with flexible instance-specific optimization.

- Authors have provided experiments demonstrating robust performance of TACO on Minimum Vertex Cover and Maximum Clique problems.

**Additional Comments:**

NA

**Audience:**

Yes

**Audience Explanation:**

Unsupervised neural combinatorial optimization is a highly active research area.

**Claims And Evidence:**

No

**Claims Explanation:**

While the submission provides empirical results that support some of its claims, several key assertions, particularly those concerning the role of the perturbation mechanism and its ability to escape poor local minima, are not supported by sufficiently rigorous theoretical analysis or convincing empirical evidence. In addition, the reported performance improvements over baselines are often marginal, making it difficult to conclusively attribute the gains to the proposed method. Overall, the evidence supports the feasibility of the approach, but it falls short of fully substantiating the stronger claims made in the paper.

**Requested Changes:**

Questions and Comments:

1. Page 4 (Perturbation in adapted parameter initialization).
   The manuscript states that “the perturbation term introduces … trapped in poor local minima.” This is a strong claim that currently lacks theoretical justification. As I understand it, the motivation is that plain gradient descent may get stuck in local minima, and the added perturbation helps escape such regions. However, this idea is not new; for instance, Perturbed Gradient Descent (PGD) has been studied extensively and enjoys theoretical guarantees for escaping saddle points [1]. Do the authors have any theoretical or empirical evidence that their proposed perturbation mechanism indeed facilitates escaping local minima or saddle points?

2. Page 5 (Synthetic dataset and data splitting).
   The authors mention constructing a synthetic dataset and splitting it into training and test sets. How many training epochs were used? Given that data are generated synthetically, it is unclear why a fixed train/test split is necessary. In streaming or on-the-fly data generation settings, it is often preferable to sample fresh data continuously, and do livestreaming training to reduce overfitting to specific configurations and ensure coverage of the full data distribution. Please explain me more about this choice.

3. Experimental results.
   In cases where TACO outperforms competing methods, the performance gains appear to be quite marginal. It would be helpful to discuss whether these improvements are statistically significant and whether they justify the added methodological complexity.

4. Overall contribution.
   The core contribution appears to be a relatively minor methodological extension that builds heavily on prior work. While this is not inherently problematic, it is currently unclear what the main conceptual or technical innovation is. The authors would benefit from more clearly articulating what is genuinely new and how it advances the state of the art beyond existing approaches.

Writing and Presentation Suggestions

5. Page 2.
   “Graph Neural Network (GNN)” should be written as “graph neural network (GNN)” to be consistent with standard abbreviation conventions used elsewhere in the paper.

6. Page 3.
   “Model-Agnostic Meta-Learning (MAML)” should be written as “model-agnostic meta-learning (MAML)” for consistency.

7. Page 5.
   “Minimum Vertex Cover (MVC)” should follow the same lowercase convention.

8. Page 5.
   “Maximum Clique (MC)” should follow the same lowercase convention.

9. Several citations are listed only as arXiv preprints. For works that have appeared in conferences or journals, please cite the official published versions rather than the arXiv versions. This is important for maintaining citation quality and formality in the review process.

[1] How to Escape Saddle Points Efficiently, Chi Jin, Rong Ge, Praneeth Netrapalli, Sham M. Kakade, Michael I. Jordan Proceedings of the 34th International Conference on Machine Learning, PMLR 70:1724-1732, 2017.

---

> ### Author Response · Authors · 2026-02-12
>
> We thank the reviewer for their thoughtful feedback. The manuscript has been revised accordingly, with all modifications highlighted in blue. Below, we address each of the concerns in detail.
>
> **(1) On escaping local minima**
>
> We agree that our work does not provide a formal theoretical guarantee for escaping local minima or saddle points. We have removed the escaping local minima claim in the noted sentence to avoid overstating.
>
> Our evidence is empirical and based on optimization behavior. In Figures 3 and 4, we observe that TACO-enhanced models consistently exhibit more favorable adaptation trajectories than naive fine-tuning: (i) in cases where fine-tuning starts to plateau, TACO continues to improve (e.g., RB200-MVC and RB500-MVC in Figure 3; RB200-MC and RB500-MC in Figure 4); (ii) when both trajectories flatten, TACO converges to better solutions (e.g., TWITTER-MC in Figure 3; TWITTER-MVC in Figure 4); and (iii) in almost all settings, TACO exhibits larger improvement slopes over the same number of update steps. With the newly added experiments on the maximum independence set problem suggested by reviewer r3t2 (Figures 6 and 7 in Appendix B.5 in the updated manuscript), these points are reinforced.
>
> These patterns are consistent with the interpretation that the shrink-and-perturb initialization moves the parameters into regions of the loss landscape that are more amenable to instance-wise optimization. While this behavior is related in spirit to perturbed optimization methods (e.g., PGD), our contribution is empirical: demonstrating that such shrink and perturb operations resolve a practical incompatibility between generalization-trained unsupervised NCO models and instance-specific optimization.
>
> **(2) Synthetic dataset generation and fixed train/test splits**
>
> We appreciate this question. Although the synthetic RB datasets are generated procedurally, we follow a fixed train/validation/test split for two reasons:
> 1. Consistency with prior work. Our setup follows Wang & Li (2023), ensuring direct comparison with EGN and Meta-EGN under identical conditions.
> 2. Controlled evaluation of adaptation. Fixing the test set allows us to isolate the effects of test-time adaptation, rather than conflating them with changes (though marginal) in instance sampling.
>
> While online or streaming data generation is indeed a valid alternative, and potentially beneficial, we intentionally used fixed train/test splits here to ensure reproducibility and comparability. In terms of training epochs, we followed Wang & Li (2023) by training the models until convergence (i.e., when the validation performance stops improving) with a limit of 1000-2000 epochs.
>
> **(3) Marginal performance gains and statistical significance**
>
> We agree that in some settings, the absolute performance gains appear numerically small. However, two points are important:
> 1. Some problems are easy. Some benchmarks (e.g., COLLAB and Twitter) are close to optimal. However, on hard instances (RB500), TACO can bring more than 1% absolute improvement over fine-tuning.
> 2. Gains are consistent and systematic. TACO consistently improves mean performance across datasets, tasks, update budgets, and backbones.
>
> Moreover, TACO often achieves in fewer update steps what baselines cannot reach even with longer optimization (see Figures 3, 4, 6, and 7), which is particularly relevant in test-time adaptation scenarios.
>
> **(4) Clarifying the core contribution**
>
> While TACO is lightweight, its novelty lies in reframing test-time adaptation for unsupervised NCO as a warm-start incompatibility problem and showing that:
> - Generalization-trained unsupervised NCO models can be worse starting points for instance-wise optimization than random initialization.
> - A simple but principled warm-start transformation (shrink + perturb) is sufficient to bridge generalization-based and instance-specific paradigms
>
> To our knowledge, this failure mode and its resolution have not been identified or empirically demonstrated in prior unsupervised NCO work. TACO is therefore a conceptual bridge that unifies two previously disjoint paradigms with minimal overhead and broad applicability.

---

> > ### Comment · Reviewer_Ps7m · 2026-02-15
> >
> > I would like to thank the authors for taking the time to address my questions.
> >
> > - Regarding the discussion on escaping local minima: do you have any experimental results comparing standard gradient descent (GD) with your proposed approach that incorporates perturbations? In particular, it would be helpful to see empirical evidence demonstrating that the perturbation mechanism indeed improves the ability to escape poor local minima or leads to better final solutions.
> >
> > - I believe the experimental section could be further strengthened by including comparisons with additional baselines, such as [1, 2], as well as other related works that you already cite (e.g., in Section 4 and Section 4.1 on datasets, particularly in the paragraph on static problems). Including these comparisons would provide a more comprehensive and convincing evaluation of your method.
> >
> > [1] Coexpander: Adaptive solution expansion for combinatorial optimization. J Ma, W Pan, Y Li, J Yan. ICML (2025).
> >
> > [2] Sebastian Sanokowski, Sepp Hochreiter, and Sebastian Lehner. A diffusion model framework for unsupervised
> > neural combinatorial optimization. ICML (2024).

---

> > > ### Author Response · Authors · 2026-02-16
> > >
> > > We thank the reviewer for the comment and suggestion.
> > >
> > > - Our baselines include standard fine-tuning, and across our experiments, TACO consistently improves final performance compared to the baseline. If by “GD” the reviewer meant the SGD optimizer (we used Adam), prior work [1] suggests that the benefits of perturbation are not specific to Adam in supervised learning settings. While we do not have experiments using SGD, previous works on unsupervised combinatorial optimization use Adam [2, 3], and we expect similar relative improvements with SGD. Adam is used here as it is the default choice in practice.
> > > - Our experimental focus is on demonstrating the relative improvement provided by TACO over the corresponding fine-tuning baseline across backbones, rather than achieving state-of-the-art performance. While we could increase model capacity of the backbones to compare with and compete against additional baselines, this is orthogonal to our main goal and would primarily improve absolute performance rather than the relative gains that TACO provides.
> > >
> > > [1] Jordan Ash and Ryan P Adams. On warm-starting neural network training. Advances in neural information processing systems, 33:3884–3894, 2020.
> > >
> > > [2] Nikolaos Karalias and Andreas Loukas. Erdos goes neural: an unsupervised learning framework for combinatorial optimization on graphs. Advances in Neural Information Processing Systems, 33:6659–6672, 2020
> > >
> > > [3] Haoyu Peter Wang and Pan Li. Unsupervised learning for combinatorial optimization needs meta learning. In The Eleventh International Conference on Learning Representations, 2023.

---

> > > > ### Comment · Reviewer_Ps7m · 2026-02-17
> > > >
> > > > 1. Thank you for the clarification. By “gradient descent,” I was referring more generally to optimizers such as SGD or Adam. I may have misunderstood part of your setup, so I would appreciate some additional clarification. You mentioned that Adam is used as the optimizer, while your method also introduces perturbations. Could you please explain how the perturbation is incorporated into training? Specifically, is it applied within the Adam update step, or as a separate modification to the parameters/gradients outside the optimizer? I want to better understand whether the perturbation is integrated into the optimizer or implemented as an additional step.
> > > >
> > > > 2. Regarding the dataset choice, I am also curious about generalization. Do you expect TACO to provide similar relative improvements on other datasets, or was this dataset selected because the gains are particularly strong there? From my perspective, the individual components (e.g., perturbation or warm-start strategies) seem related to existing ideas, so it would help to better understand what novel aspect of your approach drives the improvements and what specific gap it addresses.

---

> > > > > ### Author Response · Authors · 2026-02-17
> > > > >
> > > > > **How TACO is applied.** Our method is applied at test time instead of during training, offering plug-and-play integration. The shrink-and-perturb operation is applied as a separate parameter transformation prior to the adaptation updates. Concretely, given trained parameters $\theta$, we initialize the adapted parameters as $\theta^* \leftarrow \lambda_{\text{shrink}} \theta + \lambda_{\text{perturb}} \epsilon$. After this transformation, standard gradient updates are performed on $\theta^*$ using the unsupervised objective. The perturbation therefore does not modify the internal update rule of the optimizer; rather, it changes the starting point of instance-specific optimization.
> > > > >
> > > > > **On dataset choice and generalization.** The datasets used in our experiments follow standard benchmarks in prior unsupervised NCO work (e.g., Karalias & Loukas, 2020; Wang & Li, 2023). They were not selected based on the magnitude of TACO’s gains. Importantly, we observe consistent relative improvements across structurally different graph families (random regular graphs, RB graphs, and real-world graphs) and across different combinatorial optimization problems (MVC, MC, and MIS). This consistency suggests that the improvement is not dataset-specific but instead tied to the adaptation mechanism.
> > > > >
> > > > > **On the novel aspect and the specific gap addressed.** Our work identifies a previously unexplored phenomenon: generalization-trained unsupervised NCO models can serve as worse starting points for instance-wise optimization than even random initialization. Parameters optimized for distribution-level performance may lie in regions of the loss landscape that are stable for generalization but poorly suited for rapid per-instance refinement, leading to slow progress or early plateaus under naive fine-tuning. The core contribution of TACO is to bridge this mismatch between the generalization-based training paradigm and the instance-specific optimization paradigm. We show that a simple but principled warm-start transformation, shrink and perturb (SP), is sufficient to relocate the initialization into regions that are more amenable to instance-wise optimization. While SP was originally proposed in the supervised learning setting to mitigate generalization gaps caused by naive warm-starting during training, TACO adapts this idea to a fundamentally different regime: test-time adaptation of unsupervised neural combinatorial optimization models. In this setting, the challenge is not improving training generalization, but resolving the incompatibility between distribution-level training and instance-specific optimization. Importantly, the novelty does not lie in proposing SP individually, but in identifying this gap in unsupervised NCO and demonstrating that a structured SP initialization resolves it.

---

### Review · Reviewer_HxSD · 2026-02-04

**Summary Of Contributions:**

I have made a summary in my review for Paper6901.

**Audience:**

Yes

**Audience Explanation:**

Yes, it is an important problem addressed by a novel methodology.

**Broader Impact Concerns:**

There is no evident concern regarding the impact of the paper.

**Claims And Evidence:**

Yes

**Claims Explanation:**

Most of the claims are supported by the experimental section, with the exception of my aforementioned concerns.

**Requested Changes:**

To my unederstanding, taking the best colution out of many seeds means that a method that occasionally finds a very good solution but is unstable can look strong.

TACO-enhanced models do not surpass Gurobi in solution quality, although they are faster.

The test-time adaptation introduces additional hyperparameters (shrink and perturb coefficients), whose selection is only briefly discussed. While a sensitivity analysis is provided, it remains unclear how these parameters would be chosen in practice depending on the availability of validation data.

---

> ### Comment · Reviewer_HxSD · 2026-02-05
> **Review for Paper6901**
>
> This paper studies unsupervised neural combinatorial optimization (NCO) and addresses the gap between two dominant paradigms: generalization-focused models trained across instances, and models optimized independently at test time for each instance. The authors identify that naively fine-tuning a pretrained unsupervised NCO model at test time can perform worse than optimizing the model for that instance starting from random initialization.
>
> To resolve this, they propose TACO, an effective model-agnostic test-time adaptation framework for unsupervised NCO. TACO partially relaxes pretrained parameters to preserve learned inductive bias while enabling instance-wise adaptation. To be specific, TACO applies a structured shrink-and-perturb warm start to pretrained model parameters, followed by a small number of unsupervised gradient updates at test time. This initialization enables more flexible adaptation than naive fine-tuning while avoiding the poor local optima often encountered by both purely generalizable and purely instance-specific methods.
>
> The approach is evaluated on Minimum Vertex Cover and Maximum Clique using existing unsupervised NCO backbones, under static, distribution-shifted, and dynamic settings. The experiments explicitly evaluate models using multiple random input initializations, following the standards ofthe literature. The results indicate that TACO consistently improves solution quality over both naive fine-tuning and purely instance-specific optimization, with negligible additional computational cost, and achieves competitive performance relative to classical heuristics, sampling-based methods, and commercial solvers.
>
> The work overall is well positioned and its benefits are clear.
>
> My two minor concerns are:
>
> To my unederstanding, taking the best colution out of many seeds means that a method that occasionally finds a very good solution but is unstable can look strong.
>
> TACO-enhanced models do not surpass Gurobi in solution quality, although they are faster.
>
> The test-time adaptation introduces additional hyperparameters (shrink and perturb coefficients), whose selection is only briefly discussed. While a sensitivity analysis is provided, it remains unclear how these parameters would be chosen in practice depending on the availability of validation data.

---

> > ### Author Response · Authors · 2026-02-12
> >
> > We thank the reviewer for their thoughtful feedback. The manuscript has been revised accordingly, with all modifications highlighted in blue. Below, we address each of the concerns in detail.
> >
> > **(1) On reporting the best solution over multiple random seeds**
> >
> > We agree that reporting the best solution across multiple random seeds can, in principle, favor methods that are unstable but occasionally produce very good outcomes. Our evaluation protocol follows established practice in prior unsupervised NCO work (e.g., Karalias & Loukas, 2020; Wang & Li, 2023), where multiple seeds are used, and the best-performing seed is reported to account for the inherent stochasticity in input feature generation.
> >
> > To address this concern more directly, we additionally evaluate performance averaged over 8 random seeds. These results are now included in the revised manuscript (Appendix B.4) and below in this comment. When considering average performance, TACO consistently improves upon the corresponding fine-tuning baselines in all settings, with the exception of RB500 in the max clique experiments. This indicates that the gains from TACO are not driven by rare favorable runs but reflect systematic improvements in solution quality.
> >
> > MVC ($\downarrow$):
> > | Method | Twitter | COLLAB | RB200 | RB200 |
> > | --- | --- | --- | --- |  --- |
> > EGN (best)          | $1.03026_{\pm 0.02123}$ | $1.00071_{\pm 0.00442}$ | $1.02243_{\pm 0.00495}$ | $1.01630_{\pm 0.00208}$ |
> > EGN-TACO (best)     | $1.01668_{\pm 0.01636}$ | $1.00040_{\pm 0.00341}$ | $1.02052_{\pm 0.00420}$ | $1.01577_{\pm 0.00214}$ |
> > EGN (avg)           | $1.06519_{\pm 0.02900}$ | $1.00920_{\pm 0.02128}$ | $1.03466_{\pm 0.00359}$ | $1.02169_{\pm 0.00216}$ |
> > EGN-TACO (avg)      | $1.04388_{\pm 0.02885}$ | $1.00852_{\pm 0.02034}$ | $1.03385_{\pm 0.00329}$ | $1.02136_{\pm 0.00209}$ |
> >
> > MC ($\uparrow$):
> > | Method | Twitter | COLLAB | RB200 | RB200 |
> > | --- | --- | --- | --- |  --- |
> > EGN (best)          | $0.99151_{\pm 0.04802}$ | $0.99916_{\pm 0.01238}$ | $0.95659_{\pm 0.09698}$ | $0.95306_{\pm 0.11117}$ |
> > EGN-TACO (best)     | $0.99766_{\pm 0.01310}$ | $0.99976_{\pm 0.00580}$ | $0.98094_{\pm 0.05496}$ | $0.97101_{\pm 0.07950}$ |
> > EGN (avg)           | $0.89725_{\pm 0.09001}$ | $0.90530_{\pm 0.12583}$ | $0.78436_{\pm 0.11440}$ | $0.81894_{\pm 0.17828}$ |
> > EGN-TACO (avg)      | $0.90888_{\pm 0.07366}$ | $0.95183_{\pm 0.08028}$ | $0.79625_{\pm 0.10480}$ | $0.81644_{\pm 0.17891}$ |
> >
> > **(2) Comparison with Gurobi**
> >
> > We agree with the reviewer that TACO-enhanced models do not surpass Gurobi in solution quality. Our goal, however, is not to compete with exact solvers in optimality, but to study the trade-off between solution quality, adaptability, and runtime in settings where exact solvers are impractical. As shown in Tables 3 and 4, Gurobi achieves better solutions but often requires substantially longer runtimes on hard instances (e.g., RB500). In contrast, TACO produces high-quality solutions within practical time budgets
> >
> > **(3) Practical selection of shrink and perturb hyperparameters**
> >
> > We appreciate this question and agree that clearer guidance on hyperparameter selection is important. In practice, the shrink and perturb coefficients can be tuned once on a small validation set and then reused across test instances, as done in our experiments.
> >
> > When validation data is unavailable, our sensitivity analysis indicates that conservative default values perform robustly across datasets and problem settings. In particular, TACO is not overly sensitive to these hyperparameters; a broad range of shrink and perturb values consistently yields improvements over naive fine-tuning. This robustness reduces the burden of careful hyperparameter tuning in practical deployment.

---

### Decision · Action_Editor_pQpo · 2026-03-18

**Recommendation:** Accept with minor revision

**Additional Comments:**

The reviewers find this work is overall well positioned with clear benefits, the experimental design is described in sufficient detail and is satisfactory, and the experimental results do support the main claim of the paper. However, they also raise important concerns regarding the methodological contribution, coefficient selection, theoretical or empirical evidence supporting the claim of escaping local minima, method details, experimental setting, performance, and the need for further analysis. The authors have provided a concise response addressing these concerns with new results.

After the rebuttal, this work received mixed ratings (2 leaning accept and 1 reject). **Reviewer HxSD** thinks most of their concerns have been answered and believes the results in this work could be useful to some readers, and hence leans towards accepting this work. **Reviewer r3t2** suggests that the pre-experiment sections could be shortened to allow for more comprehensive testing of the proposed method using different GNNs and learning frameworks. On the other hand, this reviewer also finds that the paper provides sufficient evidence for its claims and would be of interest to TMLR readers working on learning-to-optimize, and thus leans toward acceptance. **Reviewer Ps7m** still has concerns regarding the methodological contribution and the insufficient experimental analysis. They believe some key claims, such as the role of perturbations in escaping local minima, are still not theoretically justified or empirically demonstrated in a convincing manner. In addition, this reviewer is also concerned about the relatively limited experimental comparison and evaluation against prior approaches, which makes it difficult to properly assess the method's added value over existing work. Consequently, this reviewer recommends rejection.

I have read this paper in detail and agree with the reviewers that the results and findings reported in this work could be of interest to some TMLR readers. At the same time, I also recognize the importance of the remaining concerns, particularly regarding the need for a more comprehensive experimental analysis to better support the claims and contributions of the work. Taking into account all review comments, the authors' rebuttal, and the reviewers' official recommendations, I recommend **accepting this work, but with a "minor" (somewhat like a major) revision**.

To further address the concerns raised by the reviewers and improve the quality of this work, I have the following suggestions.

1. Conduct further experimental analysis to address the remaining concerns raised by Reviewer Ps7m, particularly regarding the role of perturbations in escaping local minima. In addition, please include more experimental comparisons and evaluations against prior approaches.

2. Following Reviewer r3t2's valuable suggestion, consider shortening the pre-experiment sections and include more comprehensive testing of the proposed method using different GNNs and learning frameworks.

3. Given the analytical nature of this work, please open-source the code, trained models, and experiment pipeline to facilitate reproducibility.

In the camera-ready version, I expect the authors to properly address all the remaining concerns outlined above.

**Audience:**

Yes

**Audience Explanation:**

All reviewers believe some individuals in TMLR's audience could be interested in the findings of this paper.

**Claims And Evidence:**

Yes

**Claims Explanation:**

This work proposes TACO, a model-agnostic test-time adaptation framework for unsupervised combinatorial optimization. Motivated by the observation that generalization-focused model training can lead to poor initialization for instance-specific test-time adaptation, TACO introduces a simple yet effective warm-start approach. This approach computes a weighted sum of the trained model parameters and a randomly sampled perturbation term, using carefully selected coefficients. In this way, TACO encourages exploration while retaining the inductive bias learned by the trained model. Experimental results show that TACO achieves promising performance on Minimum Vertex Cover and Maximum Clique instances from various distributions.

Two reviewers believe the claims made in this paper are supported by convincing and clear evidence, but one reviewer does not think so. Please refer to the comment below for more details.

---

> ### Author Response · Authors · 2026-04-22
>
> We thank the Action Editor and the reviewers for their careful evaluation of our manuscript and for their constructive and insightful comments. We have revised the paper accordingly and summarize the main changes below.
>
> **Escaping local minima.**
> We acknowledge that perturbed gradient descent (PGD) is highly relevant to better highlight the role of perturbations, as pointed out by reviewer Ps7m. We have now explored PGD as an alternative for injecting “noise”. We include the study in Appendix B.6. We found that PGD can help reach better solutions than standard fine-tuning, but our method always yields better outcomes and can be combined with PGD to achieve additional improvements over PGD alone. In addition, as noted in our response during the rebuttal, our results are empirical, and therefore to avoid overemphasis on (analytical) support for escaping local minima, we have changed the wording in a few spots in our paper. With the revised wording, we want to highlight that our method facilitates *reaching better solutions faster* and potentially reaching solutions that are not reachable by standard fine-tuning with the max number of update steps we considered in our experiments.
>
> **Comparison against prior approaches.**
> Following the suggestion of reviewer Ps7m, we have incorporated a comparison against the state-of-the-art method COExpander. The corresponding results are reported in the Experiments section. We find that our method achieves slightly inferior yet broadly comparable solutions on min vertex cover and consistently outperforms COExpander on max clique. Also, COExpander does not consistently benefit from increased runtime, but TACO exhibits monotonic improvement as additional compute is allocated.
>
>
> **Shortening pre-experiment sections and additional backbone.**
> We have revised and shortened the pre-experiment sections to improve conciseness and clarity (keeping the overall paper length unchanged after the addition of the new experiments). In addition, as suggested by reviewer r3t2, we include an additional backbone, ConsFormer, with results reported in the Experiments section. In contrast to the other two GNN-based backbones, ConsFormer is a transformer-based architecture. We also extend our evaluation to the max cut problem. By incorporating a different class of models and an additional CO problem, we further verify the effectiveness of and generality of our method even on transformer-based architectures.
>
>
> **Code.**
> We have made our code publicly available and included a link to the repository in the Experiments section.

---

> > ### Comment · Reviewer_r3t2 · 2026-04-22
> >
> > Thank you for continuing to improve the paper based on our and the AE's recommendations -- looks great!

---

> > ### Comment · Action_Editor_pQpo · 2026-04-23
> >
> > Thank you for the careful revision. This work has now been officially accepted by TMLR.